# StateMask: Explaining Deep Reinforcement Learning through State Mask

**Zelei Cheng**[*]
Northwestern University
Evanston, IL 60208
zelei.cheng@northwestern.edu

**Xian Wu**[*]
Northwestern University
Evanston, IL 60208
xianwu2024@u.northwestern.edu

**Jiahao Yu**[*]
Northwestern University
Evanston, IL 60208
jiahao.yu@northwestern.edu

**Wenhai Sun**
Purdue University
West Lafayette, IN 47907
sun841@purdue.edu

**Wenbo Guo**
Purdue University
West Lafayette, IN 47907
henrygwb@purdue.edu

**Xinyu Xing**
Northwestern University
Evanston, IL 60208
xinyu.xing@northwestern.edu

## Abstract

Despite the promising performance of deep reinforcement learning (DRL) agents in many challenging scenarios, the black-box nature of these agents greatly limits their applications in critical domains. Prior research has proposed several explanation techniques to understand the deep learning-based policies in RL. Most existing methods explain why an agent takes individual actions rather than pinpointing the critical steps to its final reward. To fill this gap, we propose `StateMask`, a novel method to identify the states most critical to the agent's final reward. The high-level idea of `StateMask` is to learn a mask net that blinds a target agent and forces it to take random actions at some steps without compromising the agent's performance. Through careful design, we can theoretically ensure that the masked agent performs similarly to the original agent. We evaluate `StateMask` in various popular RL environments and show its superiority over existing explainers in explanation fidelity. We also show that `StateMask` has better utilities, such as launching adversarial attacks and patching policy errors.

## 1 Introduction

Deep reinforcement learning has shown promising performance in a variety of challenging scenarios, such as allowing a robot to learn a control policy directly from the camera or sensor inputs [1, 2, 3, 4, 5] or training an agent to master Go games [6, 7, 8]. However, DRL policies typically lack explainability, which is intrinsically difficult for humans to comprehend and establish trust. To broaden the adoption of DRL-based applications in critical fields, there is a pressing need to increase the transparency of DRL agents through explanation.

Prior research has proposed various methods to derive an explanation for DRL. For example, researchers utilized the saliency map (*e.g.,* [9, 10, 11, 12, 13]) and model approximation methods

---

[*]Equal Contribution.

37th Conference on Neural Information Processing Systems (NeurIPS 2023).

(*e.g.,* [14, 15, 16, 17]) to identify which regions in an agent's observation are most critical to the agent's current action. Considering the nature of DRL, researchers also explored the relationship between the agent's states and its final reward. Take the work [18] as an example. Researchers demonstrated that the state-reward relationship could also be used as an explanation for DRL agent developers. Besides, they also showed that pinpointing the critical time steps towards the final reward could help agent developers better understand agent behaviors and even troubleshoot policy defects.

In this work, our research focuses on the DRL explanation methods that explore the state-reward relationships. We evaluate the existing DRL explanation approaches in this category on a range of tasks from simple simulated environments to sophisticated real-world applications, and from normal-form games to extensive-form games. We discovered that existing state-reward-based DRL explanation methods provide relatively low fidelity in state-reward relationship identification for various reasons. For example, the method proposed in [18] suffers from low explanation fidelity because of the limited capacity of their proposed approximation model and the error introduced in the final reward approximation. The methods introduced by [19] and [20] experience low fidelity because an agent's value function does not always reflect the state importance to its final reward.

To this end, we propose `StateMask`[2], a novel method to pinpoint the time steps or, in other words, states most critical to a DRL agent's expected total reward. At a high level, `StateMask` works as follows. For a target agent, we train a mask net that blinds the agent to take random actions at certain time steps. If a time step is critical, the mask net does not blind the target agent. The agent takes action as its policy suggests. In contrast, if the time step is less critical, the mask net blinds the agent. The agent takes random actions. The rationale behind this design is that a DRL agent's final reward is not equally important for each time step. Randomly taking some actions at non-critical time steps has no impact on the agent's final reward. Technically, to learn the mask net, we carefully design an objective function that minimizes the performance difference between the original agent and the masked agent. We demonstrate, both theoretically and empirically, that performance differences could monotonically decrease during training.

While other explanation methods could track down the time steps most or least critical to an agent's final reward, this is the first work that could identify an agent's critical time steps with high fidelity across various types of RL environments, especially sophisticated normal-form games and extensive-form games. Evaluating `StateMask` in various game environments, we show that the explanation derived from our method demonstrates significantly higher fidelity than those produced by existing explanation techniques. Compared with the existing methods, `StateMask` allows humans to understand the agent's behaviors better and calibrate appropriate trust. Agent users can easily pinpoint an agent's policy defects and patch the identified defeat under the guidance of explanations. Last but not least, security professionals can launch adversarial attacks using the explanation from our method.

In summary, this paper makes the following contributions.

- We propose a new explanation method – `StateMask` and provide a theoretical analysis. We show that an agent armed with explainability is equivalent to the original agent in terms of agent performance.

- We evaluate `StateMask` on 10 different tasks ranging from simple normal-form games (*e.g.,* Pong) to sophisticated normal-form games (*e.g.,* StarCraft II) and extensive-form games (*e.g.,* Connect 4). We show that `StateMask` can significantly outperform existing explainers.

- We demonstrate that our explanation can be highly helpful for understanding agent behavior, launching adversarial attacks, and even remediating the agent's errors.

## 2    Related Work

**DRL Explanation.** Existing DRL explanation techniques can be mainly categorized into two classes based on their different explanation objectives: (1) the observation-action-based method pinpointing which regions in an agent's observation are most important to its action at an individual time step; 2) the state-reward-based method tracking down which time steps have the most critical influence on the agent's final reward.

---

[2]The source code of `StateMask` can be found in `https://github.com/nuwuxian/RL-state_mask`.

Recall that a DRL agent's policy network takes as input the agent's current observation and outputs the corresponding action. In this setup, pinpointing regions in an observation most critical to the corresponding action is the same as identifying the most important regions in a deep neural network's (DNN) input. As such, existing works along this direction mainly utilize or extend the explanation methods developed for DNN classifiers to derive such explanations. Technically, they mainly leverage three lines of methods: post-hoc white-box explanation methods [21, 22, 23, 24], post-hoc black-box explanation methods [15, 25, 26, 27], and self-explainable methods [28, 29, 30, 31]. In addition, another line of work identifies important regions by decomposing the agent's instant reward to regions in its current observation and selecting the regions with high rewards [32, 33, 34, 35].

Regarding the second type of explanation methods, most of the techniques identify the critical steps based on the value function [19, 20, 36]. Given that the value function measures the goodness of the current state towards the agent's expected total reward, it is natural to use it for critical time step selection. Recently, Guo *et al.* [18] proposed EDGE that establishes state-reward relationship by collecting a set of trajectories and then approximating an explanation model offline. However, as discussed in [18], EDGE has two limitations. First, the approximation method is based on a statistical model not effective in deriving explanations for agents that usually generate long trajectories. Second, for each new trajectory, EDGE needs to build an independent approximation model, which significantly limits its efficiency and scalability.

Our proposed technique falls into the second type of explanation, which follows a different design path to overcome the limitations above. As we will discuss later, our method is superior to existing techniques in multiple aspects: higher explanation fidelity (Section 4), better utilities (Section 5), as well as better generalizability (to extensive-form games) and practicability (in the cases where the agent's value function/network is not available).

**Perturbation-based Explanation.** Perturbation techniques have been widely utilized within the field of computer vision to gain a deeper understanding of black-box image classifier [37, 38, 39, 40]. These approaches involve introducing minor modifications to the target image and observing how the classifier's outputs change, which enables us to identify the regions of greatest importance within the original image. Our explanation method shares a similar idea with these perturbation methods. Rather than explaining supervised learning (*e.g.,* image classifier), our focus is on interpreting a DRL agent by introducing perturbations in non-critical time steps. This temporal perspective provides unique insights into the DRL agent, which has not been explored in prior research.

## 3 Key Technique

In the following, we describe the rationales behind our design and then discuss the key technical challenges, followed by the technical details of tackling the challenges.

### 3.1 Overview

**Assumptions.** To ensure the practicability of our method, we do not assume access to the agent's value function or policy network. Instead, we only assume having access to the agent's observations and the ability to query the agent's policy and thereby obtain corresponding actions. Note that our explanation method does not change the target agent's policy. For a multi-agent environment, we explain each agent individually.

**Design Rationale.** To tackle the limitations confronted by existing state-reward-based explanation methods, our idea is to parameterize the importance of the agent's current time step as a neural network model (denoted as state mask). The state mask net takes as input the current state and outputs its importance to the agent's final reward. The *key challenge* of this solution is to propose a proper criterion or metric to decide the importance of the current time step and thus design an objective function to train this state mask. In our work, we draw insights from existing counterfactual-based explanation methods designed for supervised classifiers [37, 41, 42, 43]. At a high level, these methods identify important features in an input sample by nullifying or perturbing the features and observing the corresponding changes in the classification output. They deem the features triggering the largest prediction changes as the most important ones. Similarly, in our problem, we can decide whether a time step is important by randomizing the agent's action at a certain time step and observing changes in its final reward. A small variation in the final reward indicates the reward is not sensitive

 

(a) The DRL agent just launched the ball.      (b) The DRL agent attends to catch the ball.

Figure 1: Illustration of the Pong game. The DRL agent controls the yellow (right) paddle and a non-RL rule-based opponent controls the white (left) paddle.

to the action change, further implying the low importance of the state. In contrast, a large reward change infers the high importance of the corresponding state.

To elaborate on this idea, we use a pong game as an example. As shown in Figure 1(a), after the DRL agent launches the ball and when the ball is moving towards its opponent, the agent has a few time steps of freedom to choose random actions. This is because the agent's actions at these steps will not change the movement of the ball and will be less likely to influence the opponent's actions as well (given the opponent's focus is on the ball). As a result, forcing the DRL agent to take random actions at these steps will likely have no influence on the game's final result and thus trigger only minor changes to the agent's final reward. However, if we randomly perturb the DRL agent's action at the time steps when the agent is about to touch the ball (Figure 1(b)), it will likely cause the agent to fail and thus trigger a large variation in the agent's final reward.

**State Mask Modeling.** Guided by this idea, we design the state mask that takes the current state as input and outputs the probability of whether to randomize the agent's action at the current state. The objective of this state mask is to blind a target agent and thus vary its actions at certain states in a trajectory without changing the agent's final reward. This probability can be taken as the importance of the current state (*i.e.,* the higher the probability is, the lower the importance is). This design introduces *a challenge*. That is how to decide whether varying the action at the current state will influence the agent's future reward.

To address this challenge, we model it as an RL problem, where we fix the target agent's policy and treat it as part of the environment. Then, we introduce another agent to decide whether to randomize the target agent's action at each step or not. Its policy can be modeled as the state mask above. This agent's goal is to preserve the target agent's total reward while randomizing actions at more states.

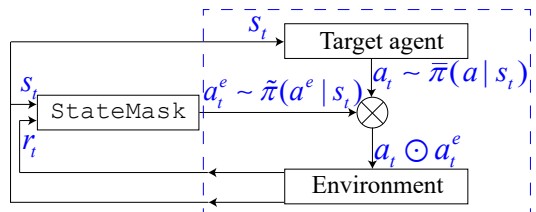

Figure 2: The `StateMask` framework.

Following this idea, we propose a novel explanation framework, `StateMask`. Figure 2 illustrates this explanation system. As mentioned above, we add a state mask (agent) to the environment. We fit the current state $s_t$ into both the state mask $\tilde{\pi}$ and the target agent $\bar{\pi}$. For a given state $s_t$, the state mask outputs a binary action $a_t^e$ of either "zero" or "one", and the target agent will sample the action $a_t$ from its policy. The final action is determined by the following equation

$$a_t \odot a_t^e = \begin{cases} a_t, & \text{if } a_t^e = 0\,, \\ a_{\text{random}} & \text{if } a_t^e = 1\,, \end{cases} \tag{1}$$

where $a_t^e = 0$ represents keeping the target agent's current action. On the contrary, $a_t^e = 1$ refers to replacing the target agent's current action with a random action $a_{\text{random}}$ uniformly sampled from its action space. Note that $a_{\text{random}}$ is different from $a_t$.

The goal of our state mask is to find the non-important time steps and randomize their actions without changing the expected total reward of the target agent. Formally, it can be expressed as [3]

$$J(\theta) = \text{minimize} |\eta(\pi) - \eta(\bar{\pi})|\,, \tag{2}$$

where $\bar{\pi}$ denotes the policy of the target agent and $\pi$ denotes the policy of the perturbed agent (*i.e.,* integration of the state mask $\tilde{\pi}$ and target agent $\bar{\pi}$). $\eta(\cdot)$ is the expected total reward of an agent by following a certain policy. By fixing $\bar{\pi}$, $\eta(\bar{\pi})$ is also fixed and $\eta(\pi)$ depends only on the parameters

---

[3]We measure the performance gap by the $L_p$ distance, and without loss of generality, we use the $L_1$ norm.

of the state mask. As a result, by solving Eqn. (2), we can obtain a state mask. By applying this resolved mask to each state, we will be able to assess the state importance at any time step.

**Key Optimization Challenges and Solutions.** Obtaining a non-trivial solution of Eqn. (2) presents the following challenge. To ensure the optimization function is differentiable, we need to decompose and remove the absolute value operation. A straightforward solution is to compare the value of $\eta(\pi)$ and $\eta(\bar{\pi})$ during each training iteration. If $\eta(\pi)$ is larger than $\eta(\bar{\pi})$, we update the state mask by minimizing $\eta(\pi)$. Otherwise, we maximize $\eta(\pi)$. With simple reformulation, we can then leverage the state-of-art policy gradient method – Proximal Policy Optimization (PPO) [44] to update the state mask in each iteration. However, the key limitation of this solution is the lack of convergence guarantee. Suppose the policy of the perturbed agent $\pi$ is near a local optimal and $\eta(\pi)$ is less than $\eta(\bar{\pi})$. Then, updating the perturbed policy by maximizing $\eta(\pi)$ may lead to an overly large $\eta(\pi)$, which, in turn, requires minimizing $\eta(\pi)$. When this situation occurs, the perturbed policy $\pi$ may oscillate around the local optimal without actually converging. To address this problem, we borrow the idea of TRPO technique [45], carefully designing a surrogate objective function for Eqn. (2). As we will elaborate below, through the surrogate objective function, we can guarantee the difference between $\eta(\pi)$ and $\eta(\bar{\pi})$ (*i.e.,* $|\eta(\pi) - \eta(\bar{\pi})|$) monotonically decreases during the training process, and our optimization process will at least converge to a local optimal rather than oscillating around it.

## 3.2 Technical Details

In this section, we formally introduce our proposed `StateMask` framework, including modeling the state mask and designing the surrogate objective function to ensure monotonicity.

**Defining the State Mask.** An infinite-horizon Markov Decision Process (MDP) is defined as $\{\mathcal{S}, \mathcal{A}, P, R, \gamma\}$, where $\mathcal{S}$ and $\mathcal{A}$ are the state and action space. $P : \mathcal{S} \times \mathcal{A} \to \Delta(\mathcal{S})$ is the state transition function. $R : \mathcal{S} \times \mathcal{A} \to \mathbb{R}$ is the reward function and $\gamma \in (0, 1)$ is the discount factor. A policy $\pi$ is a mapping from $\mathcal{S}$ to a probability distribution over $\mathcal{A}$. The expected total reward of a policy $\pi$ is defined as $\eta(\pi) = \mathbb{E}_{s_0,a_0,\dots} \left[ \sum_{t=0}^{\infty} \gamma^t R(s_t, a_t) \right]$.

Based on this definition, we define our explanation as an Explanation Markov Decision Process problem (E-MDP), in which we formalize state mask as an explanation policy $\tilde{\pi} : \mathcal{S} \to \Delta(\mathcal{A}^e)$, where $\mathcal{A}^e = \{0, 1\}$. With this definition, the integration of the explanation policy $\tilde{\pi}$ and the target policy $\bar{\pi}$ (that we aim to explain) then becomes $\pi(\cdot|s) = \tilde{\pi}(a^e = 0|s)\bar{\pi}(\cdot|s) + \tilde{\pi}(a^e = 1|s)a_{\text{random}}$, where $a_{\text{random}} \neq \bar{\pi}(\cdot|s)$.

As mentioned above, we fix $\bar{\pi}$. In this way, we can transform the E-MDP into a single-player environment, where the state-transition, reward function, and $\eta(\pi)$ depend only on the policy of the state mask $\tilde{\pi}$. As such, we can redefine $\eta(\pi)$ as $\eta(\tilde{\pi})$. The goal here is to train an optimal policy $\tilde{\pi}$ to minimize the difference between $\eta(\bar{\pi})$ and $\eta(\tilde{\pi})$.

**Solving the Explanation Policy.** With the definitions above, the objective function in Eqn. (2) could be reformulated as $J(\theta) = \text{minimize}|\eta(\tilde{\pi}_\theta) - \eta(\bar{\pi})|$, where $\theta$ refers to the parameters of the explanation policy. As is discussed in Section 3.1, to ensure the monotonic property, we subtly design a surrogate objective function to solve $\theta$. To introduce our designed objective function, we first present the following lemma.

**Lemma 1.** *([45]) The difference between $L_{\tilde{\pi}_{\theta_{old}}}(\tilde{\pi}_\theta)$ and $\eta(\tilde{\pi}_\theta)$ is bounded by*

$$|\eta(\tilde{\pi}_\theta) - L_{\tilde{\pi}_{\theta_{old}}}(\tilde{\pi}_\theta)| \le C\mathbb{KL}^{max}(\tilde{\pi}_{\theta_{old}}(\cdot \mid s)\|\tilde{\pi}_\theta(\cdot \mid s)). \tag{3}$$

$\tilde{\pi}_\theta$ denotes the new policy, updated from $\tilde{\pi}_{\theta_{old}}$. $L_{\tilde{\pi}_{\theta_{old}}}(\tilde{\pi}_\theta) = \eta(\tilde{\pi}_{\theta_{old}}) + \sum_s \rho_{\tilde{\pi}_{\theta_{old}}}(s) \sum_{a^e} \tilde{\pi}_\theta(a^e \mid s) A_{\tilde{\pi}_{\theta_{old}}}(s, a^e)$, where $\rho_{\tilde{\pi}_{\theta_{old}}}(s) = \sum_t \gamma^t P(s_t = s|\tilde{\pi}_{\theta_{old}})$ and $A_{\tilde{\pi}_{\theta_{old}}}$ is the advantage function of $\tilde{\pi}_{\theta_{old}}$. $\mathbb{KL}^{max}(\tilde{\pi}_{\theta_{old}}(\cdot \mid s)\|\tilde{\pi}_\theta(\cdot \mid s)) = \max_s \mathbb{KL}(\tilde{\pi}_{\theta_{old}}(\cdot \mid s)\|\tilde{\pi}_\theta(\cdot \mid s))$ and $C$ is a constant.

Based on Lemma 1, we can derive the following theorem (The derivation is in Supplement S1).

**Theorem 1.** *The following inequalities are the sufficient conditions for satisfying $|\eta(\tilde{\pi}_\theta) - \eta(\bar{\pi})| \le |\eta(\tilde{\pi}_{old}) - \eta(\bar{\pi})|$.*

$$
\begin{aligned}
&L_{\tilde{\pi}_{\theta_{old}}}(\tilde{\pi}_\theta) \ge \eta(\tilde{\pi}_{\theta_{old}}) + C\mathbb{KL}^{max}(\tilde{\pi}_{\theta_{old}}(\cdot \mid s)\|\tilde{\pi}_\theta(\cdot \mid s)), \\
&L_{\tilde{\pi}_{\theta_{old}}}(\tilde{\pi}_\theta) + C\mathbb{KL}^{max}(\tilde{\pi}_{\theta_{old}}(\cdot \mid s)\|\tilde{\pi}_\theta(\cdot \mid s)) \le 2\eta(\bar{\pi}) - \eta(\tilde{\pi}_{\theta_{old}}).
\end{aligned}
\tag{4}
$$

**Algorithm 1** The learning algorithm for training the mask net.

---

**Input:** Target agent's policy $\bar{\pi}$ and the estimation of its expected total reward $\eta(\bar{\pi})$
**Output:** Mask net $\tilde{\pi}_\theta$
**Initialization:** Initialize the weights $\theta$ for the mask net $\tilde{\pi}_\theta$, Lagrange multiplier $\lambda = 0$
**for** iteration=1, 2, ... **do**
    Run the current policy $\tilde{\pi}_{\theta_{old}}$ in the environment for $T$ steps and get estimation for $A_{\tilde{\pi}_{\theta_{old}}}$
    Solve the objective function with respect to $\theta$ in Eqn. (7) and update $\theta_{old} \leftarrow \theta$
    Solve the objective function with respect to $\lambda$ in Eqn. (8) and update $\lambda_{old} \leftarrow \lambda$
**end for**

---

Based on Theorem 1, we propose the following objective function to update $\tilde{\pi}_\theta$ from $\tilde{\pi}_{\theta_{old}}$.

$$
\begin{aligned}
\max_\theta \quad & L_{\tilde{\pi}_{\theta_{old}}}(\tilde{\pi}_\theta) \\
\text{s.t.} \quad & L_{\tilde{\pi}_{\theta_{old}}}(\tilde{\pi}_\theta) \leq 2\eta(\bar{\pi}) - \eta(\tilde{\pi}_{\theta_{old}}) - \delta, \\
& \mathbb{E}_{a^e \sim \tilde{\pi}_\theta}[a^e] \geq c, \quad \mathbb{E}_{s \sim \rho_{\tilde{\pi}_{\theta_{old}}}}[\mathbb{KL}(\tilde{\pi}_{\theta_{old}}(\cdot \mid s) \| \tilde{\pi}_\theta(\cdot \mid s))] \leq \delta.
\end{aligned}
\tag{5}
$$

where $\mathbb{E}_{a^e \sim \tilde{\pi}_\theta}[a^e] \geq c$ constrains the lower bound sparsity for the output of $\tilde{\pi}_\theta$ across all states, with $c \in [0, 1]$ as a constant hyper-parameter. It rules out the trivial solution where the state mask outputs 0 at any state. As detailed in Supplement S2, a policy $\tilde{\pi}_\theta$ solved from Eqn. (5) satisfies the conditions in Eqn. (4) and thus enables the desired monotonicity.

Eqn. (5) can be further transformed into the following function based on the Lagrangian duality.

$$
\begin{aligned}
\min_{\lambda \geq 0} \max_\theta \quad & L_{\tilde{\pi}_{\theta_{old}}}(\tilde{\pi}_\theta) - \lambda(L_{\tilde{\pi}_{\theta_{old}}}(\tilde{\pi}_\theta) - 2\eta(\bar{\pi}) + \eta(\tilde{\pi}_{\theta_{old}}) + \delta), \\
\text{s.t.} \quad & \mathbb{E}_{a^e \sim \tilde{\pi}_\theta}[a^e] \geq c, \quad \mathbb{E}_{s \sim \rho_{\tilde{\pi}_{\theta_{old}}}}[\mathbb{KL}(\tilde{\pi}_{\theta_{old}} \| \tilde{\pi}_\theta)] \leq \delta.
\end{aligned}
\tag{6}
$$

Below we further discuss our techniques for solving Eqn. (6). In particular, we first follow the PPO algorithm [44] and transform the Eqn. (6) into the following objective function for $\theta$ (See Supplement S3 for more details)

$$
\max_\theta \mathbb{E}\left[ \text{sgn}(1 - \lambda) \min\left( \frac{\tilde{\pi}_\theta(a_t^e \mid s_t)}{\tilde{\pi}_{\theta_{old}}(a_t^e \mid s_t)} A_{\tilde{\pi}_{\theta_{old}}}, \text{clip}\left( \frac{\tilde{\pi}_\theta(a_t^e \mid s_t)}{\tilde{\pi}_{\theta_{old}}(a_t^e \mid s_t)}, 1 - \epsilon, 1 + \epsilon \right) A_{\tilde{\pi}_{\theta_{old}}} \right) + w L_t^{\text{MASK}} \right].
\tag{7}
$$

Here, $A_{\tilde{\pi}_{\theta_{old}}} = A_{\tilde{\pi}_{\theta_{old}}}(s_t, a_t^e)$ and $L_t^{\text{MASK}} = \mathbb{E}_{a_t^e \sim \tilde{\pi}_\theta}[a_t^e]$. sgn is the sign function which returns 1 or $-1$ based on whether the input is positive or negative. $w$ is a hyperparameter controlling the weight of the corresponding term during optimization. The outer expectation is taken over a finite batch of state and action, sampled by running the current policy $\tilde{\pi}_{\theta_{old}}$.

Similarly, we can also transform Eqn. (6) into the following objective function with respect to $\lambda$

$$
\min_{\lambda \geq 0} \lambda(2\eta(\bar{\pi}) - \eta(\tilde{\pi}_{\theta_{old}}) - \delta - L_{\tilde{\pi}_{\theta_{old}}}(\tilde{\pi}_\theta)).
\tag{8}
$$

We follow the primal-dual method [46, 47] and iteratively update $\theta$ and $\lambda$ by solving Eqn. (7) and Eqn. (8) using the gradient-based optimization method. Algorithm 1 briefly presents our learning process. Supplement S4 shows a more detailed algorithm including how to estimate $\eta(\bar{\pi})$ and $A_{\tilde{\pi}_{\theta_{old}}}$.

As mentioned above, as an explanation method, we fix the policy of the target agent in the environment and only train our state mask. For a multi-agent environment, we can learn an individual state mask for each target agent. This ensures the generalizability of our method to multi-agent setups. As specified in Supplement S4, with minor implementation modifications, we can also generalize our method from normal-form games, where the agents observe the same state and take action simultaneously, to extensive-form agents, where the agents make decisions in sequential order.

## 4 Evaluation

In the following, we start with our experiment setup and design, followed by experiment results and analysis. To ensure the reproducibility of our results, we specify the evaluation details in Supplement S5.

## 4.1 Experiment Setup

**Environment Selection.** We select *10* representative environments to demonstrate the effectiveness of `StateMask` across four types of environments: simple normal-form game (CartPole, Pendulum, and Pong [48]), sophisticated normal-form game (You-Shall-Not-Pass [49], Kick-And-Defend [49], and StarCraft II [50]), perfect-information (simple) extensive-form game (Connect 4, Tic-Tac-Toe and Breakthrough [51]), and imperfect-information (sophisticated) extensive-form game (DouDizhu [52]).

**Baseline Selection.** Sharing the same explanation goal, as discussed in Section 2, two existing methods explore the state-reward relationship – the value-function-based method and EDGE [18]. For the value-function-based method, we implement two methods - Value-max method [19, 20] and LazyMDP [36]. The value-max method associates the state importance with its value function $V(s)$. LazyMDP pinpoints critical steps according to the lazy gap, which is defined as $\max_a Q(s, a) -$ $\mathbb{E}_a[Q(s, a)]$. Here, $Q(s, a)$ is the action-value function. For EDGE, we implement it based on their code [18]. We take an absolute value for the important scores drawn by these methods to align their meaning with our methods.

## 4.2 Experiment Design

**Experiment I: Expected Reward Preservation.** Recall the state mask blinds the agent and forces it to take random actions at some least important time steps. If the blinding is accurate, the agent's performance should not be affected, and the agent's final reward should not vary significantly. As a result, one of our evaluations is to examine whether the masked agent could preserve its expected total reward. To do so, for a target agent from a game, we compute its discounted total reward across 500 rounds. Then, we treat the averaged computation result as the estimation of the agent's expected total reward $\eta(\bar{\pi})$. Following the computation of the expected reward, we further train our `StateMask` in the corresponding game.

As mentioned above, the combination of the state mask and the target agent's policy is the perturbed policy. Therefore, after having the state mask in hand, we further compute the discounted total reward of the perturbed policy across 500 rounds. We treat the averaged computation result as the approximation of $\eta(\tilde{\pi}_\theta)$, and then use it to compute the relative value variation $\frac{|\eta(\bar{\pi}) - \eta(\tilde{\pi}_\theta)|}{|\eta(\bar{\pi})|}$. As is shown from this equation, the relative value variation represents the change of the reward. If this measure is low, it indicates that the state mask has a minimal impact upon the target agent's performance. The state mask accurately pinpoints the time step least important to the final reward.

**Experiment II: Explanation Fidelity.** Apart from evaluating our explanation method based on the measure of expected reward preservation, we also employ a fidelity test proposed in [18] to assess its effectiveness. The fidelity test involves using the corresponding explanation method to identify and rank the most crucial time steps in a given trajectory. Subsequently, a continuous sequence of time steps is selected based on the rank of the time steps. The fidelity test evaluation algorithm proposed in [18] ensures that the selected sequence includes the maximum number of critical steps, indicating the most crucial continuous time steps leading to the final reward of the agent. Additionally, the length of the sequence can be specified as an input to the fidelity test evaluation algorithm. For instance, one can limit the sequence to only the top $K$ time steps. For more information about the fidelity test, readers can refer to Supplement S5.

Following the rest of the fidelity test introduced in [18], for a given trajectory and the most critical sequence in it, we let the agent fast-forward to the beginning of the critical sequence. Then, we force the target agent to take random actions until the end of the critical sequence. Later onwards, we use the target agent's policy to complete the game. By doing so, we simulate a situation where we force the agent not to follow its policy at the most critical time steps. If the time steps are truly critical to the final reward, we expect the action replacement at critical time steps could introduce the highest reward variation compared with replacing any other continuous sequence containing $K$ time steps. In [18], Guo *et al.* introduce a fidelity score to measure the change of the reward after the replacement of the actions at a continuous sequence of time steps. The higher the measure is, the higher the explanation fidelity is.

In this work, we use `StateMask` (the state masks trained above) and baseline methods to explain 500 trajectories of each target agent and obtain the time step importance for each trajectory. Then, for each trajectory, we select their top $K$ important steps based on the importance ranking given by each

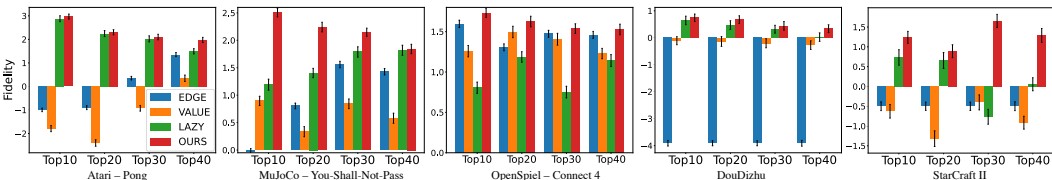

Figure 3: Fidelity scores of explanations generated by selected methods in 5 games. The x-axis represents the choice of $K$. "VALUE" stands for Value-max method, and "LAZY" stands for LazyMDP. The black line refers to the standard deviation of 3 runs. The bar refers to the mean and the line on the bar represents the standard deviation.

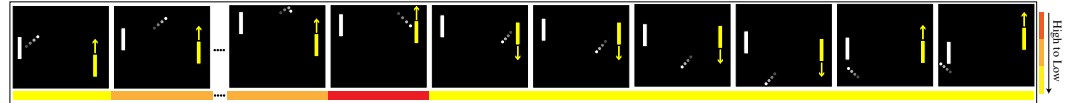

Figure 4: Visualization of the identified critical time steps in Pong game. The DRL agent controls the yellow paddle and a non-RL rule-based opponent controls the white paddle.

method and compute the fidelity score. We compute the average score across 500 trajectories as the fidelity score for that explanation method. We set $K = 10\%, 20\%, 30\%, 40\%$ and report the fidelity of the selected methods under each setup. We repeat each experiment 3 times with different random seeds and report the mean and standard deviation.

## 4.3 Experiment Result

Due to the space limit, we show the results of five games: Pong, You-Shall-Not-Pass, Connect 4, DouDizhu, and StarCraft II. Supplement S5 shows the other games' results, which are aligned with those discussed in this section.

**Relative Value Errors.** The maximum/minimum/mean relative value errors across all 10 games are 14.98%/0.00%/5.17% (The exact values are in Supplement S5). These results verify the effectiveness of our design objective function (Eqn (6)) in preserving the target agent's performance. In addition, we also show the trends and variations in $|\eta(\bar{\pi}) - \eta(\tilde{\pi}_\theta)|$ during the training process for each game. The figures in Supplement S5 demonstrate that our design could indeed achieve the monotonical property, and further improve the stability of the training process.

**Fidelity Scores.** Figure 3 shows the fidelity scores of `StateMask` against the three baseline approaches in five games while the results of the remaining five games could be found in Supplement S5. First, we can observe that our method has the highest fidelity scores across all games in all settings, indicating that `StateMask` provides more faithful explanations for the target agents than the baseline approaches. Second, we discover that in two complex normal-form and extensive-form games (DouDizhu and StarCraft II), `StateMask` has a significant advantage over EDGE. This result validates our argument in Section 2 that EDGE's approximation fails to handle long trajectories. Besides, we discover that `StateMask` outperforms value-based methods in those two games. We believe that the reason is that the value function is trained to facilitate policy learning and typically tends to output similar values for a sequence of continuous states [36], causing the time step importance to be less distinguishable. While `StateMask` specifically quantifies the importance of each time step via action randomization, giving more specific and distinguishable important scores.

In addition to the experiments above, we also demonstrate our method's efficiency and its insensitivity against the variations in the key hyper-parameters. We also verify the effectiveness of our method on not well-trained policies with sub-optimal performance. Due to the space limit, we detail these experiments in the supplementary materials. We further conduct an additional comparison to verify the necessity of minimizing the difference in the agent's expected total reward before and after applying the state mask. We consider an alternative design that directly maximizes the agent's expected total reward after applying the state mask (*i.e.,* $J(\theta) = \max_\theta \eta(\tilde{\pi}_\theta)$) and compares it with our method. Our results show that `StateMask` produces higher fidelity explanations on not well-trained policies than this alternative method (See Supplement S10 for more details).

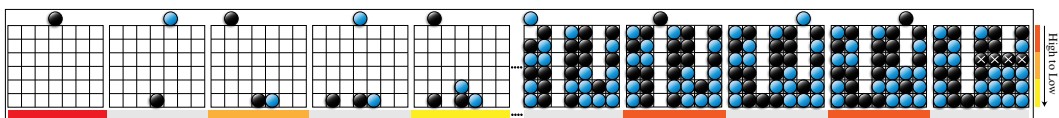

Figure 5: Visualization of the identified critical steps in the Connect 4 game. The black player is our explanation target player. The states where the blue player takes actions are marked with a gray bar.

Table 1: The target agent's performance in different applications. Numbers before the brackets are means and those in the brackets are standard deviations.

| Applications | Games | EDGE | Value-max | LazyMDP | Ours |
|---|---|---|---|---|---|
| Target agent's winning rate change before/after adversarial attacks | Pong | -63.04 (3.44) | -0.40 (0.40) | -36.17 (1.15) | **-84.90 (1.40)** |
| | You-Shall-Not-Pass | -2.64 (0.60) | -14.80 (3.20) | -28.87 (1.21) | **-35.93 (1.81)** |
| | Connect 4 | -33.33 (1.56) | -26.30 (1.44) | -22.57 (1.40) | **-35.37 (3.00)** |
| | DouDizhu | -0.25 (1.71) | -18.60 (0.31) | -25.20 (0.76) | **-29.94 (0.64)** |
| | StarCraft II | -70.00 (14.18) | -77.33 (5.69) | -72.00 (13.53) | **-83.00 (7.21)** |
| Target agent's winning rate change before/after patching | Pong | +1.13 (0.12) | +0.53 (0.12) | +1.47 (0.12) | **+1.87 (0.12)** |
| | You-Shall-Not-Pass | +0.07 (0.12) | +1.07 (0.12) | +0.12 (0.12) | **+2.42 (0.18)** |
| | Connect 4 | +2.00 (0.00) | +1.47 (0.12) | +1.33 (0.12) | **+2.40 (0.20)** |
| | DouDizhu | +0.00 (0.00) | +0.13 (0.12) | +0.20 (0.00) | **+0.47 (0.12)** |

# 5 Utility of Explanation

In this section, we demonstrate that with a higher explanation fidelity, `StateMask` could also achieve better utilities in three aspects: understanding agent behaviors, launching adversarial attacks, and patching agent errors.

**Understanding Agent Behaviors.** We visualize the identified critical steps in all 10 games to demonstrate how `StateMask` helps humans understand a DRL agent's behavior. Due to the space limit, we show the results of two games: Pong (normal-form game), Connect 4 (perfect-information extensive-form game), and leave the rest in Supplement S6.

Regarding the Pong game as Figure 4 shows, our explanation method highlights the frames when the ball is approaching the yellow paddle as the most important time steps whereas the states when the ball flies to the white paddle as the least important time steps. It aligns with humans' intuition since the steps when the ball is flying towards the yellow paddle are apparently critical to the final rewards. Without proper actions at these critical time steps, the DRL agent will miss the ball and lose the game.

In Connect 4, two players take turns dropping colored stones onto a grid to connect four stones in a line. Figure 5 visualizes one game where the black player wins. `StateMask` pinpoints the first time step (where the target agent makes the first move) as an important step. Allis *et al.* [53] shows that placing a stone in the center column as the first action typically brings an advantage. `StateMask` also highlights the last two time steps, which drive the target agent to win the game. [4]

Additionally, we conduct a user study [5] to compare our explanation model with baseline methods (Value-max, EDGE, and LazyMDP) and see which one could help humans gain a better understanding of a DRL agent's behavior. Our study is determined to be exempt from an IRB review by the IRB panel. In our user-study survey, the participants are first given some background information about this project and asked for their consent to proceed. The hypothesis to be tested is that our method can offer a better explanation of the target agent's behavior in comparison with baseline methods. We use the convenience sampling method [54] to recruit 41 participants for our study. The participants come from a variety of backgrounds in explainable reinforcement learning (XRL), with 20% having never heard of XRL, 51% having some knowledge of it, and 29% being an expert or having published papers in this field. Next, the participants are presented with two groups of videos of the Kick-And-Defend game. The videos record the agent's play and highlight the critical time steps based on different explanation methods . Additionally, these two groups include trajectories played by near-optimal and sub-optimal agents. The participants are instructed to watch these two groups of videos and choose which explanation best matched their intuition.

---

[4]In the second-to-last step, the foundation for the game's victory is established. No matter which column the blue player places their piece, the black player has a winning strategy.

[5]The link to our survey: `https://tinyurl.com/ytwj7wdt`.

According to the survey results in Figure 6, 71% of the participants prefer our explanation method when the agent is near-optimal and 59% of them favor our explanation method when the agent is sub-optimal. Since the majority of participants choose our method, our hypothesis is confirmed. Our method outperforms the baseline methods in helping the user understand a DRL agent's policy.

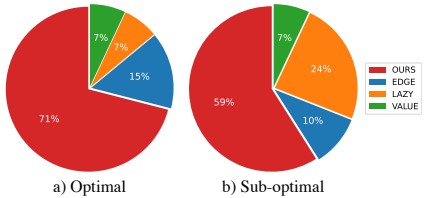

Figure 6: Results of our user study.

**Launching Adversarial Attacks.** We follow the method developed in EDGE [18] and launch attacks against the target agent based on explanations drawn by the selected methods. Specifically, we fix the percent of time steps under attack $T$ as 10% and launch this attack against 500 episodes. We then compute the average reward that the agent gathers before and after the attack. Table 1 shows the results in five games (same as the games in Section 4.3). Supplement S7 further shows additional results when varying the attack ratio from 10% to 30% in all ten games. As shown in the figure, the adversarial attack under the guidance of all explanation methods could dramatically drop the victim agent's performance, indicating the effectiveness of the interpretation and the exploitability of the attack. However, the performance losses observed on the adversarial attacks under the guidance of the three baseline methods are significantly lower than what we observed on the attack enabled by our method. This again implies that the attack guided by our method is more effective, and thus our interpretation has higher fidelity than the three baseline methods.

**Patching Agent Errors.** Rather than directly applying the rule-based patching method in EDGE [18], we propose a novel learning-based method (Supplement S8 shows the advantages of our learning-based method over the rule-based one proposed in EDGE). Given a set of trajectories, we first leverage selected explanation methods to identify a sequence of continuous time steps that are most critical to the final reward in each trajectory. Then, we replay these trajectories to the starting point of the critical steps and fine-tune the agent's policy till the game end. We validate our patching method on seven games (except CartPole, Breakthrough, and StarCraft II whose agent has a nearly optimal performance) and report the results in Table 1 and Supplement S8. As shown in the table, our method-guided agent achieves the best performance after retraining.

## 6 Conclusion and Discussion

In this work, we propose `StateMask`, a novel explanation method for DRL agents. The use of `StateMask` as an explainer can reveal the state-reward relationship with exceptional accuracy. Compared to other types of explainers, `StateMask` offers the highest level of fidelity in explanation. By providing such precise explanations, we can assist humans in comprehending agent behaviors, launching adversarial attacks, and even identifying and correcting policy errors.

Our work suggests several promising directions for future research. First, in safety-critical environments, masking the agent's state and forcing it to take random actions may not be feasible due to high exploration costs. However, to train DRL agents in such environments, the model developer typically needs to build a simulator for the environment, which can be used to generate explanations. As part of future work, we will explore building simulators for such an environment if unavailable. Second, although `StateMask` achieves high fidelity, it may not identify cases where positively and negatively important states in the same trajectory offset each other's influence on the final reward. In the future, we will investigate additional steps to handle such cases. Third, while our explanation methods can identify important states, presenting the states to humans may not be ideal. Other future efforts will also include converting critical states to human-understandable strategies. Finally, our study in Section 5 shows the feasibility of explanation-guided policy fine-tuning. We plan to improve the fine-tuning efficiency and the generalizability of the retrained DRL agents in the future.

## Acknowledgement

This project was supported in part by NSF Grant 2225234, NSF Grant 2225225, NSF Grant 2238680, and Amazon Research Award.

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

# StateMask: Explaining Deep Reinforcement Learning through State Mask

## S1 Proof of Theorem 1

Before we proving Theorem 1, we first introduce the following lemma.

**Lemma 2.** *Given an old and new policy of the state mask $\tilde{\pi}_{\theta_{old}}$ and $\tilde{\pi}_\theta$, respectively, if $\eta(\tilde{\pi}_{\theta_{old}}) \leq \eta(\tilde{\pi}_\theta) \leq 2\eta(\bar{\pi}) - \eta(\tilde{\pi}_{\theta_{old}})$, we have $|\eta(\tilde{\pi}_\theta) - \eta(\bar{\pi})| \leq |\eta(\tilde{\pi}_{\theta_{old}}) - \eta(\bar{\pi})|$.*

**Proof of Lemma 2.** The monotonicity property requires $|\eta(\tilde{\pi}_\theta) - \eta(\bar{\pi})| \leq |\eta(\tilde{\pi}_{\theta_{old}}) - \eta(\bar{\pi})|$. The following equivalence holds:

$$
\begin{aligned}
& |\eta(\tilde{\pi}_\theta) - \eta(\bar{\pi})| \leq |\eta(\tilde{\pi}_{\theta_{old}}) - \eta(\bar{\pi})| \\
\iff & \eta(\tilde{\pi}_\theta)^2 - 2\eta(\tilde{\pi}_\theta)\eta(\bar{\pi}) + \eta(\bar{\pi})^2 \leq \eta(\tilde{\pi}_{\theta_{old}})^2 - 2\eta(\tilde{\pi}_{\theta_{old}})\eta(\bar{\pi}) + \eta(\bar{\pi})^2 \\
\iff & (\eta(\tilde{\pi}_\theta) + \eta(\tilde{\pi}_{\theta_{old}}))(\eta(\tilde{\pi}_\theta) - \eta(\tilde{\pi}_{\theta_{old}})) \leq 2\eta(\bar{\pi})(\eta(\tilde{\pi}_\theta) - \eta(\tilde{\pi}_{\theta_{old}})) .
\end{aligned}
\tag{1}
$$

Therefore, if $\eta(\tilde{\pi}_{\theta_{old}}) \leq \eta(\tilde{\pi}_\theta)$ and $\eta(\tilde{\pi}_\theta) + \eta(\tilde{\pi}_{\theta_{old}}) \leq 2\eta(\bar{\pi})$, or in other words, $\eta(\tilde{\pi}_{\theta_{old}}) \leq \eta(\tilde{\pi}_\theta) \leq 2\eta(\bar{\pi}) - \eta(\tilde{\pi}_{\theta_{old}})$, we have $|\eta(\tilde{\pi}_\theta) - \eta(\bar{\pi})| \leq |\eta(\tilde{\pi}_{\theta_{old}}) - \eta(\bar{\pi})|$. $\qquad\square$

Recall that Lemma 1 in Section 3.2 shows the upper bound and the lower bound of $\eta(\tilde{\pi}_\theta)$. An immediate implication of Lemma 1 enables us to obtain the inequalities to satisfy Lemma 2.

From Lemma 1, we can easily derive the lower bound of $\eta(\tilde{\pi}_\theta)$ as $\eta(\tilde{\pi}_\theta) \geq L_{\tilde{\pi}_{\theta_{old}}}(\tilde{\pi}_\theta) - C\mathbb{KL}^{\max}(\tilde{\pi}_{\theta_{old}}(\cdot \mid s)\|\tilde{\pi}_\theta(\cdot \mid s))$. If the lower bound is greater than $\eta(\tilde{\pi}_{\theta_{old}})$, then we can guarantee that $\eta(\tilde{\pi}_\theta) \geq \eta(\tilde{\pi}_{\theta_{old}})$. Similarly, we can obtain the upper bound of $\eta(\tilde{\pi}_\theta)$ as $\eta(\tilde{\pi}_\theta) \leq L_{\tilde{\pi}_{\theta_{old}}}(\tilde{\pi}_\theta) + C\mathbb{KL}^{\max}(\tilde{\pi}_{\theta_{old}}(\cdot \mid s)\|\tilde{\pi}_\theta(\cdot \mid s))$. If the upper bound is lower than $2\eta(\bar{\pi}) - \eta(\tilde{\pi}_{\theta_{old}})$, we can guarantee that $\eta(\tilde{\pi}_\theta) \leq 2\eta(\bar{\pi}) - \eta(\tilde{\pi}_{\theta_{old}})$.

Putting all these inequalities together, the following inequalities are the sufficient conditions for satisfying $|\eta(\tilde{\pi}_\theta) - \eta(\bar{\pi})| < |\eta(\tilde{\pi}_{old}) - \eta(\bar{\pi})|$:

$$
\begin{aligned}
& L_{\tilde{\pi}_{\theta_{old}}}(\tilde{\pi}_\theta) \geq \eta(\tilde{\pi}_{\theta_{old}}) + C\mathbb{KL}^{\max}(\tilde{\pi}_{\theta_{old}}(\cdot \mid s)\|\tilde{\pi}_\theta(\cdot \mid s)) \\
& L_{\tilde{\pi}_{\theta_{old}}}(\tilde{\pi}_\theta) + C\mathbb{KL}^{\max}(\tilde{\pi}_{\theta_{old}}(\cdot \mid s)\|\tilde{\pi}_\theta(\cdot \mid s)) \leq 2\eta(\bar{\pi}) - \eta(\tilde{\pi}_{\theta_{old}}) .
\end{aligned}
\tag{2}
$$

## S2 Derivation of Eqn. (5)

First, let $M_{\tilde{\pi}_{\theta_{old}}}(\tilde{\pi}_\theta) = L_{\tilde{\pi}_{\theta_{old}}}(\tilde{\pi}_\theta) - C\mathbb{KL}^{\max}(\tilde{\pi}_{\theta_{old}}(\cdot \mid s)\|\tilde{\pi}_\theta(\cdot \mid s))$. We can get $M_{\tilde{\pi}_{\theta_{old}}}(\tilde{\pi}_{\theta_{old}}) = \eta(\tilde{\pi}_{\theta_{old}})$. By maximizing the $M_{\tilde{\pi}_{\theta_{old}}}(\tilde{\pi}_\theta)$, we can ensure $M_{\tilde{\pi}_{\theta_{old}}}(\tilde{\pi}_\theta) \geq M_{\tilde{\pi}_{\theta_{old}}}(\tilde{\pi}_{\theta_{old}}) = \eta(\tilde{\pi}_{\theta_{old}})$. Given that $\eta(\tilde{\pi}_\theta) \geq M_{\tilde{\pi}_{\theta_{old}}}(\tilde{\pi}_\theta)$, we could satisfy the first inequality of Eqn. (4).

By constraining the upper bound of $\tilde{\pi}_{\theta_{old}}$ to be smaller than $2\eta(\bar{\pi}) - \eta(\tilde{\pi}_{\theta_{old}})$, we could guarantee the second inequality of Eqn. (4) holds.

Therefore, combining all these constraints together, we can obtain the following optimization objective:

$$
\begin{aligned}
\max_\theta \quad & L_{\tilde{\pi}_{\theta_{old}}}(\tilde{\pi}_\theta) - C\mathbb{KL}^{\max}(\tilde{\pi}_{\theta_{old}}(\cdot \mid s)\|\tilde{\pi}_\theta(\cdot \mid s)) \\
\text{s.t.} \quad & L_{\tilde{\pi}_{\theta_{old}}}(\tilde{\pi}_\theta) + C\mathbb{KL}^{\max}(\tilde{\pi}_{\theta_{old}}(\cdot \mid s)\|\tilde{\pi}_\theta(\cdot \mid s)) \leq 2\eta(\bar{\pi}) - \eta(\tilde{\pi}_{\theta_{old}}) .
\end{aligned}
\tag{3}
$$

In practice, Schulman *et al.* [11] pointed out that utilizing the penalty coefficient $C$ may lead to a very small step size. Therefore, we replace the penalty on the maximal KL-divergence with a trust region constraint on an average KL-divergence (*i.e.*, $\mathbb{E}_{s \sim \rho_{\tilde{\pi}_{\theta_{old}}}}[\mathbb{KL}(\tilde{\pi}_{\theta_{old}}(\cdot \mid s)\|\tilde{\pi}_\theta(\cdot \mid s))] \leq \delta$) as suggested in [11]. We can further obtain the following objective

$$\max_{\theta} \quad L_{\tilde{\pi}_{\theta_{old}}}(\tilde{\pi}_{\theta})$$
$$\text{s.t.} \quad L_{\tilde{\pi}_{\theta_{old}}}(\tilde{\pi}_{\theta}) \leq 2\eta(\bar{\pi}) - \eta(\tilde{\pi}_{\theta_{old}}) - \delta, \tag{4}$$
$$\mathbb{E}_{s \sim \rho_{\tilde{\pi}_{\theta_{old}}}} [\mathbb{KL}(\tilde{\pi}_{\theta_{old}}(\cdot \mid s) \| \tilde{\pi}_{\theta}(\cdot \mid s))] \leq \delta.$$

Additionally, we add a sparsity constraint which will only tighten the problem without affecting the monotonicity property and obtain the following optimization objective

$$\max_{\theta} \quad L_{\tilde{\pi}_{\theta_{old}}}(\tilde{\pi}_{\theta})$$
$$\text{s.t.} \quad \textcircled{1} \quad \mathbb{E}_{a^e \sim \tilde{\pi}_{\theta}} [a^e] \geq c,$$
$$\textcircled{2} \quad L_{\tilde{\pi}_{\theta_{old}}}(\tilde{\pi}_{\theta}) \leq 2\eta(\bar{\pi}) - \eta(\tilde{\pi}_{\theta_{old}}) - \delta, \tag{5}$$
$$\textcircled{3} \quad \mathbb{E}_{s \sim \rho_{\tilde{\pi}_{\theta_{old}}}} [\mathbb{KL}(\tilde{\pi}_{\theta_{old}}(\cdot \mid s) \| \tilde{\pi}_{\theta}(\cdot \mid s))] \leq \delta,$$

where $\mathbb{E}_{a^e \sim \tilde{\pi}_{\theta}} [a^e] \geq c$ constrains the lower bound sparsity for the output of $\tilde{\pi}_{\theta}$ across all states. By maximizing the local approximation with constraint $\textcircled{3}$, we could guarantee that $L_{\tilde{\pi}_{\theta_{old}}}(\tilde{\pi}_{\theta}) \geq L_{\tilde{\pi}_{\theta_{old}}}(\tilde{\pi}_{\theta_{old}}) = \eta(\tilde{\pi}_{\theta_{old}})$. Since $L_{\tilde{\pi}_{\theta_{old}}}(\tilde{\pi}_{\theta})$ is the local approximation of $\eta(\tilde{\pi}_{\theta})$, we could further guarantee that $\eta(\tilde{\pi}_{\theta}) \geq \eta(\tilde{\pi}_{\theta_{old}})$. Moreover, the constraint $L_{\tilde{\pi}_{\theta_{old}}}(\tilde{\pi}_{\theta}) \leq 2\eta(\bar{\pi}) - \eta(\tilde{\pi}_{\theta_{old}}) - \delta$ could ensure that the inequality $\eta(\tilde{\pi}_{\theta}) \leq 2\eta(\bar{\pi}) - \eta(\tilde{\pi}_{\theta_{old}})$ is satisfied. Therefore, a policy $\tilde{\pi}_{\theta}$ solved from the optimization objective above satisfies Lemma 2 and thus enables the desired monotonicity.

## S3 Derivative from Eqn. (6) to of Eqn. (7)

With respect to $\theta$, Eqn. (6) becomes

$$\max_{\theta} \quad (1 - \lambda) L_{\tilde{\pi}_{\theta_{old}}}(\tilde{\pi}_{\theta}) + C_1 \quad \text{s.t.} \quad \mathbb{E}_{s \sim \rho_{\tilde{\pi}_{\theta_{old}}}} [\mathbb{KL}(\tilde{\pi}_{\theta_{old}}(\cdot \mid s) \| \tilde{\pi}_{\theta}(\cdot \mid s))] \leq \delta, \quad \mathbb{E}_{a^e \sim \tilde{\pi}_{\theta}} [a^e] \geq c, \tag{6}$$

where $C_1 = \lambda(2\eta(\bar{\pi}) - \eta(\tilde{\pi}_{\theta_{old}}) - \delta)$ is a constant that can be eliminated during the optimization of $\theta$. The KKT condition [1] implies that there exists Lagrange multiplier $\xi$ (dual solution) such that the set of solutions of the constrained optimization problem Eqn. (6) are equivalent to those of the following relaxed objective:

$$\max_{\theta} \quad (1 - \lambda) L_{\tilde{\pi}_{\theta_{old}}}(\tilde{\pi}_{\theta}) + \xi \mathbb{E}_{a^e \sim \tilde{\pi}_{\theta}} [a^e] \quad \text{s.t.} \quad \mathbb{E}_{s \sim \rho_{\tilde{\pi}_{\theta_{old}}}} [\mathbb{KL}(\tilde{\pi}_{\theta_{old}}(\cdot \mid s) \| \tilde{\pi}_{\theta}(\cdot \mid s))] \leq \delta. \tag{7}$$

If $\lambda < 1$, the optimization problem is equivalent to

$$\max_{\theta} \quad L_{\tilde{\pi}_{\theta_{old}}}(\tilde{\pi}_{\theta}) + w \mathbb{E}_{a^e \sim \tilde{\pi}_{\theta}} [a^e] \quad \text{s.t.} \quad \mathbb{E}_{s \sim \rho_{\tilde{\pi}_{\theta_{old}}}} [\mathbb{KL}(\tilde{\pi}_{\theta_{old}}(\cdot \mid s) \| \tilde{\pi}_{\theta}(\cdot \mid s))] \leq \delta, \tag{8}$$

where $w = \xi/(1 - \lambda)$.
The optimization problem can further be transformed as

$$\max_{\theta} \sum_{s} \rho_{\tilde{\pi}_{\theta_{old}}}(s) \sum_{a^e} \tilde{\pi}_{\theta}(a^e \mid s) A_{\tilde{\pi}_{\theta_{old}}}(s, a^e) + w \mathbb{E}_{a^e \sim \tilde{\pi}_{\theta}} [a^e]$$
$$\text{s.t.} \quad \mathbb{E}_{s \sim \rho_{\tilde{\pi}_{\theta_{old}}}} [\mathbb{KL}(\tilde{\pi}_{\theta_{old}}(\cdot \mid s) \| \tilde{\pi}_{\theta}(\cdot \mid s))] \leq \delta. \tag{9}$$

Due to the summation of the new policy $\tilde{\pi}_{\theta}$, optimization is still challenging to implement even after transformation. To resolve this issue, we follow the idea from [10]. Specifically, we use importance sampling and substitute it with the summation of the old policy through the Monte Carlo method [14]. Then, in order to reach our ultimate optimization function, we further substitute the trust region constraints with the clipped ratio operation proposed by the Proximal Policy Optimization algorithm (PPO) [11]. Therefore, if $\lambda < 1$, we could obtain the final objective to optimize $\theta$ as

$$\max_{\theta} \mathbb{E} \left[ \min \left( \frac{\tilde{\pi}_{\theta}(a_t^e \mid s_t)}{\tilde{\pi}_{\theta_{old}}(a_t^e \mid s_t)} A_{\tilde{\pi}_{\theta_{old}}}, \text{clip} \left( \frac{\tilde{\pi}_{\theta}(a_t^e \mid s_t)}{\tilde{\pi}_{\theta_{old}}(a_t^e \mid s_t)}, 1 - \epsilon, 1 + \epsilon \right) A_{\tilde{\pi}_{\theta_{old}}} \right) + w L_t^{\text{MASK}} \right], \tag{10}$$

where $L_t^{\text{MASK}} = \mathbb{E}_{a_t^e \sim \tilde{\pi}_{\theta}} [a_t^e]$ is added to ensure the sparsity of critical steps.

If $\lambda > 1$, the optimization problem is equivalent to

$$\min_{\theta} L_{\tilde{\pi}_{\theta_{old}}}(\tilde{\pi}_\theta) - w\mathbb{E}_{a^e \sim \tilde{\pi}_\theta}[a^e] \quad \text{s.t.} \quad \mathbb{E}_{s \sim \rho_{\tilde{\pi}_{\theta_{old}}}}[\mathbb{KL}(\tilde{\pi}_{\theta_{old}}(\cdot \mid s) \| \tilde{\pi}_\theta(\cdot \mid s))] \leq \delta. \tag{11}$$

Similar to the analysis above, if $\lambda > 1$, the final objective for updating $\theta$ becomes

$$\max_\theta \mathbb{E}\left[-\min\left(\frac{\tilde{\pi}_\theta(a_t^e \mid s_t)}{\tilde{\pi}_{\theta_{old}}(a_t^e \mid s_t)} A_{\tilde{\pi}_{\theta_{old}}}, \text{clip}\left(\frac{\tilde{\pi}_\theta(a_t^e \mid s_t)}{\tilde{\pi}_{\theta_{old}}(a_t^e \mid s_t)}, 1-\epsilon, 1+\epsilon\right) A_{\tilde{\pi}_{\theta_{old}}}\right) + wL_t^{\text{MASK}}\right], \tag{12}$$

Putting all these objectives together, we get the final objective for updating $\theta$ as

$$\max_\theta \mathbb{E}\left[\text{sgn}(1-\lambda)\min\left(\frac{\tilde{\pi}_\theta(a_t^e \mid s_t)}{\tilde{\pi}_{\theta_{old}}(a_t^e \mid s_t)} A_{\tilde{\pi}_{\theta_{old}}}, \text{clip}\left(\frac{\tilde{\pi}_\theta(a_t^e \mid s_t)}{\tilde{\pi}_{\theta_{old}}(a_t^e \mid s_t)}, 1-\epsilon, 1+\epsilon\right) A_{\tilde{\pi}_{\theta_{old}}}\right) + wL_t^{\text{MASK}}\right]. \tag{13}$$

# S4 Our Training Algorithm

## S4.1 Algorithm Detail

**Optimizing $\theta$.** Recall that the final objective function of $\theta$ is

$$\max_\theta \mathbb{E}\left[\text{sgn}(1-\lambda)\min\left(\frac{\tilde{\pi}_\theta(a_t^e \mid s_t)}{\tilde{\pi}_{\theta_{old}}(a_t^e \mid s_t)} A_{\tilde{\pi}_{\theta_{old}}}, \text{clip}\left(\frac{\tilde{\pi}_\theta(a_t^e \mid s_t)}{\tilde{\pi}_{\theta_{old}}(a_t^e \mid s_t)}, 1-\epsilon, 1+\epsilon\right) A_{\tilde{\pi}_{\theta_{old}}}\right) + wL_t^{\text{MASK}}\right], \tag{14}$$

where $A_{\tilde{\pi}_{\theta_{old}}} = A_{\tilde{\pi}_{\theta_{old}}}(s_t, a_t^e)$ and $L_t^{\text{MASK}}(\theta) = \mathbb{E}_{a_t^e \sim \tilde{\pi}_\theta}[a_t^e]$.
Note that the output of `StateMask` $a_t^e$ is sampled from a categorical distribution over all actions (*i.e.*, 0 or 1). However, as stated in [4], the categorical variable $a_t^e$ is not capable of backpropagating through samples. Note that $\mathbb{E}_{a_t^e \sim \tilde{\pi}_\theta}[a_t^e] = \mathbb{E}_{s_t \sim \rho_{\tilde{\pi}_\theta}}\left[\mathbb{E}_{a_t^e \sim \tilde{\pi}_\theta(\cdot|s)}[a_t^e]\right] = \mathbb{E}_{s_t \sim \rho_{\tilde{\pi}_\theta}}[Pr(a_t^e = 1)]$ and $Pr(a_t^e = 1)$ is differentiable. Therefore, we replace $\mathbb{E}_{a_t^e \sim \tilde{\pi}_\theta}[a_t^e]$ with $\mathbb{E}_{s_t \sim \rho_{\tilde{\pi}_\theta}}[Pr(a_t^e = 1)]$ in practice.
When estimating $A_{\tilde{\pi}_{\theta_{old}}}$, we adopt the same estimation formula as PPO[11]

$$\begin{aligned}
\hat{A}_{\tilde{\pi}_{\theta_{old}}} &= \delta_t + (\gamma\zeta)\delta_{t+1} + \cdots + \cdots + (\gamma\zeta)^{T-t+1}\delta_{T-1} \\
\text{where} \quad \delta_t &= r_t + \gamma V(s_{t+1}) - V(s_t)
\end{aligned} \tag{15}$$

Therefore, during the training process, in addition to updating the actor by solving $\theta$ in Eqn. (14), we also have to update the state-value function $V(s)$ by the Temporal-Difference (TD) method [13] for better estimating the advantage.
**Optimizing $\lambda$.** And recall that the objective function of $\lambda$ is

$$\min_{\lambda \geq 0} \lambda\left(2\eta(\bar{\pi}) - \eta(\tilde{\pi}_{\theta_{old}}) - \delta - L_{\tilde{\pi}_{\theta_{old}}}(\tilde{\pi}_\theta)\right). \tag{16}$$

where $L_{\tilde{\pi}_{\theta_{old}}}(\tilde{\pi}_\theta) = \sum_s \rho_{\tilde{\pi}_{\theta_{old}}}(s) \sum_{a^e} \tilde{\pi}_\theta(a^e \mid s) A_{\tilde{\pi}_{\theta_{old}}}(s, a^e)$.
In practice, $\eta(\tilde{\pi}_{\theta_{old}})$ and $\eta(\bar{\pi})$ can be approximated via Monte Carlo method [7]. $\eta(\bar{\pi})$ can be estimated in advance, *i.e.*, running the game with the agent's policy $\bar{\pi}$ 500 times and calculating the average expected discounted reward as follows

$$\eta(\bar{\pi}) = \mathbb{E}_{s_0, a_0, \ldots \sim \bar{\pi}}\left[\sum_{t=0}^{\infty} \gamma^t r(s_t)\right], \tag{17}$$

$\eta(\pi_{\theta_{old}})$ can be estimated using a similar formula in parallel with the collecting trajectories of old policy $\pi_{\theta_{old}}$ while solving $\theta$ with our training algorithm.
Nevertheless, the optimization objective remains challenging to solve because of the summation in $L_{\tilde{\pi}_{\theta_{old}}}(\tilde{\pi}_\theta)$ involving the new policy $\tilde{\pi}_\theta$. To address this issue, we utilize importance sampling and substitute the summation over the new policy $\tilde{\pi}_\theta$ with a summation over the old policy $\tilde{\pi}_{\theta_{old}}$, which can be computed using the Monte Carlo method [7]. Mathematically, $L_{\tilde{\pi}_{\theta_{old}}}(\tilde{\pi}_\theta)$ can be estimated by $\frac{\tilde{\pi}_\theta(a^e|s)}{\tilde{\pi}_{\theta_{old}}(a^e|s)} A_{\tilde{\pi}_{\theta_{old}}}(s, a^e)$, which has already been computed in Eqn. (14) when updating $\theta$.

**Algorithm 1** The training algorithm of mask net.
***
**Input:** Target agent's policy $\bar{\pi}$
**Output:** Mask net policy $\tilde{\pi}$
**Initialization:** Initialize the weights $\theta$ for the mask net $\tilde{\pi}$, Lagrange multiplier $\lambda = 0$
Estimate the $\eta(\bar{\pi})$ in advance by Eqn. (17)
**for** $iteration = 1, 2, \dots$ **do**
    $s \leftarrow s_0$
    $\mathcal{D} \leftarrow \emptyset$
    **for** $t = 1, 2, ..., T$ **do**
        $a_t^e \leftarrow$ sample from $\tilde{\pi}_{\theta_{old}}(\cdot|s)$
        $a_t \leftarrow$ sample from $\bar{\pi}(\cdot|s)$
        Compute actual taken action $a \leftarrow a_t \odot a_t^e$
        Obtain current step reward $r_t$, next state $s'$ from the environment given action $a$
        $\mathcal{D} \leftarrow \mathcal{D} \cup (s, a_t^e, r_t, s')$
        $s \leftarrow s'$
    **end for**
    Compute estimates for $A_{\tilde{\pi}_{\theta_{old}}}(s_t, a_t^e)$ using samples in $\mathcal{D}$ by Eqn. (15)
    Solve $\theta$ using the trajectory samples in $\mathcal{D}$ by Eqn. (14) and update $\theta_{old} \leftarrow \theta$
    Estimate the $\eta(\tilde{\pi})$ using the trajectory samples in $\mathcal{D}$ by Eqn. (17)
    Solve $\lambda$ by Eqn. (16) and update $\lambda_{old} \leftarrow \lambda$
**end for**
***

**Extensive-form Games.** Regarding extensive-form games, we train a state mask for the target agent while keeping all the agents' policies fixed. In this way, we could transform the extensive-form games into single-player normal-form games (*i.e.,* we treat all of the agents' actions as part of the environment). However, the game may not end with the target agent, and the target agent will not receive the game reward in this case. To address this problem, we automatically overwrite the target agent's final reward with the final game result.

Based on what we have discussed above, we present the full algorithm in Algorithm 1.

## S4.2 Implementation and Hyperparameters

We implement our training algorithm using PyTorch [9] and additionally discuss the hyperparameter setting of our training algorithm. The common hyperparameter setting is as follows. Regarding the clipping parameter $\epsilon$ in Eqn. (14), we set it as 0.2. As for the advantage estimation in Eqn. (15), we set $\gamma = 0.99$ and $\zeta = 0.95$. Moreover, we provide the game-specific hyperparameter choices in Table 1. Regarding the policy network of `StateMask`, we utilize a widely used network structure, commonly employed to train Deep Reinforcement Learning (DRL) agents for each game. Specifically, we use Convolutional Neural Network (CNN) for the Pong game and Residual Network (ResNet) for the Connect 4 game since these two games' input observation contains spatial information and are relatively complicated. For other games, we use either Long Short-Term Memory (LSTM) or Multi-Layer Perceptron (MLP) network, which depends on the input type of the observation and the game's complexity. We use two separate networks with the same backbone architecture for the policy and value networks. The output dimension of the policy network and value network are 2 and 1, respectively. As for the learning rate setting, we refer to other work of training a reinforcement learning agent in these games [6, 2]. Moreover, our method introduces an extra hyperparameter – $w$ (To encourage the mask network to blind the target agent at some states, we add a sparsity constraint on the number of mask actions). We search the $w$ in {0, 1e-5, 1e-4, 1e-3, 1e-2} and report the sensitivity experiment results in Supplement S5.3.

Table 1: Training hyperparameter settings of our method. The numbers in the bracket after "CNN" represent the number of kernels in each layer. The numbers in the bracket after "MLP" are the hidden dimensions. $512 \times 10$ represents 10-layer MLP with hidden size 512.

| Games | $w$ | Learning rate | Optimizer | `StateMask` backbone | Batch Size | Num. of Training Steps |
|---|---|---|---|---|---|---|
| Pong | 1e-4 | 1e-5 | Adam | CNN(16, 32) | 32 | 2e+4 |
| You-Shall-Not-Pass | 0 | 3e-4 | Adam | MLP(256, 256) | 2048 | 2e+7 |
| Kick-And-Defend | 0 | 3e-4 | Adam | LSTM(512, 256) | 2048 | 2e+7 |
| CartPole | 0 | 1e-3 | Adam | MLP(256, 256) | 5 | 8e+2 |
| Pendulum | 0 | 1e-3 | Adam | MLP(256, 256) | 5 | 8e+2 |
| StarCraft II | 0 | 1e-6 | Adam | MLP(128,128,128) | 32 | 8e+7 |
| Connect 4 | 1e-3 | 1e-3 | Adam | ResNet-128 | 1024 | 4e+5 |
| Tic-Tac-Toe | 1e-5 | 1e-2 | Adam | MLP(128, 128) | 128 | 2e+4 |
| Breakthrough | 1e-3 | 1e-3 | Adam | MLP($512 \times 10$) | 1024 | 2e+6 |
| DouDizhu | 1e-3 | 3e-4 | Adam | MLP($512 \times 10$) | 42 | 8e+5 |

# S5   Details of Evaluation

## S5.1   Other Implementation Details

**Baseline Implementations.** Regarding baseline approaches, we use the code released by the authors or implement our own version if the authors don't release the code. Specifically, as for EDGE, we use their released open-sourced code from `https://github.com/Henrygwb/edge`. For the Value-max and the LazyMDP method, we assume that we have access to the target agent's value function and implement them based on the description of the original method[1].

**Fidelity Tests and Metrics.** As mentioned in Section 4.2, we adopt the experiment design in EDGE [3] to access the accuracy of the explanation. Specifically, the fidelity test first gathers $N$ (*i.e.,* $N$ = 500 in our setting) trajectories. Each explanation method generates the importance score of each state accordingly without altering the agent's action. For each trajectory, we identify and record the length $l$ of the longest continuous sequence within the top-$K$ most critical time steps. Following this, we replay the sequence of actions from the original trajectory up to the start of this identified sequence. From this point onwards, we replace the original actions with randomly selected actions for the next $l$ steps. Once the action substitution is completed, we present the newly reached state to the target agent at the end of the selected time steps. From this point forward, the agent operates according to the actions suggested by its policy network until the conclusion of the game. The reward difference before and after action replacement is denoted as $d$. The fidelity score is defined as $\log(p_d) - \log(p_l)$ where $p_d = |d|/d_{max}$ is the absolute reward difference divided by the maximum possible reward difference and $p_l = l/L$ is the length of the longest sequence normalized by the original trajectory length $L$. Note that in the original definition of the fidelity score [3], a lower score implies a higher level of fidelity. However, for ease of presentation, we have introduced a negative sign to the fidelity score, such that a higher score indicates superior fidelity.

## S5.2   Experiment Setup

**Game Introduction and Target Agents.** To assist in determining whether our explanation aligns with human perception, we briefly introduce each game below.

- **Pong** is a two-dimensional sports game that simulates table tennis as Figure 1 shows. The RL agent controls the right paddle while a decent AI from the Atari game environment controls the left paddle. The RL agent observes itself, the opponent, and the position of the ball before deciding whether to move the paddle up, down, or take no action. In each round, the RL agent wins the game and collects a +1 reward if the opponent fails to return the ball; the agent loses the game and collects a 0 reward if it misses the ball. To facilitate the evaluation, we terminate the game at the end of each round.

---

[1]For the Value-max, since it is relatively easy to implement, we only need to extract the state value from the value function. As to the LazyMDP, we contacted the author and implemented the explainer based on the author's response.

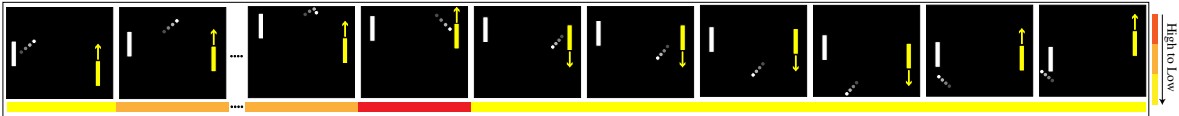

Figure 1: Visualization of the identified critical time steps in Pong game. The DRL agent controls the yellow paddle and a non-RL rule-based opponent controls the white paddle.

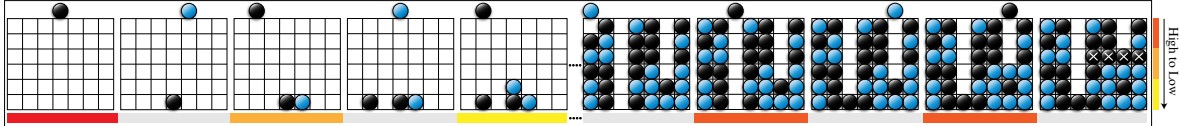

Figure 2: Visualization of the identified critical steps in the Connect 4 game. The black player is our explanation target player. Note that the states where the blue player takes actions are marked with a gray bar.

- **You-Shall-Not-Pass** is a two-party competitive game as Figure 6 shows. Two humanoid robots face each other at the beginning of the game. The red robot then starts running towards the finish line (indicated by the red line), while the blue robot is tasked with preventing the runner from achieving its goal. In our evaluation, the red robot wins the game if it can get across the finish line. Otherwise, the blue blocker wins. The red robot gets the reward +1 once it wins the game and -1 when it loses the game

- **Kick-And-Defend** is a two-party competitive game as Figure 7 shows. Two humanoid robots compete in a soccer penalty shootout. The placements of kicker, defender, and ball are all chosen at random. If the ball falls between the goalposts, the kicker wins; otherwise, the defender wins. The reward is +1 for the kicker to win the game and -1 when the kicker loses.

- **CartPole** is one of OpenAI's open-source environments as Figure 8 shows. The "cartpole" agent is a reverse pendulum in which a "cart" attempts to vertically balance a "pole" with a slight angle adjustment. The only forces that can be used are represented by +1 and -1, corresponding to a left and right movement, respectively. The episode ends if the cart moves more than 2.4 units away from the center or if the angle deviates from vertical by 15 degrees. For each timestamp that the episode hasn't ended, the reward is increased by one. The maximum reward available is 500.

- **Pendulum** simulates the inverted pendulum swingup problem as Figure 9 shows. The pendulum starts in a random position, and the goal is to swing it up and keep it upright. The pendulum angle is represented by $\theta$, which starts at random from $-\pi$ to $\pi$. The default reward function is based on the angle $\theta$ and the agent's action $a$. To maximize rewards, the agent must keep the angle $\theta$ at zero, rotate at the slowest possible speed, and exert the least amount of effort.

- **StarCraft II** is a popular real-time strategy game as Figure 10 shows. The player controls an army to defeat the opponent's base by collecting resources and building units. The player's observation is the minimap and the screen. The minimap shows the entire map, while the screen shows the player's view. The player can take actions such as selecting units, and issuing commands to the units. The player wins if the opponent's base is destroyed, and the game ends if the player's base is destroyed. The winning, and losing outcomes of the player correspond to +1 and -1 rewards, respectively.

- **Connect 4** uses a $6 \times 7$ grid as Figure 2 shows. Players take turns dropping tokens into the grid. The tokens land straight down, taking up the lowest available space in the column. The player wins if any four of the tokens are in a row horizontally, vertically, or diagonally. The game could also end in a draw if the grid is full while no player wins. The winning, draw, and losing outcomes of the player correspond to +1, 0, and -1 rewards, respectively.

- **Tic-Tac-Toe** uses a $3 \times 3$ grid as Figure 11 shows. Two players iteratively place a symbol on an available position. The player wins if any three of the symbols are in a row horizontally, vertically, or

Table 2: Game information with the target agents to explain. "Discrete(X)" refers to X discrete actions available to choose from. "Vector(X)" refers to X-dimension continuous action space.

| Games | Observation shape | Action space | Reward range | Target agent |
|---|---|---|---|---|
| Pong | (84,84) | Discrete(6) | 0,1 | Pong agent |
| You-Shall-Not-Pass | (380) | Vector(17) | -1,1 | Runner |
| Kick-and-Defend | (380) | Vector(17) | -1,1 | Kicker |
| CartPole | (4) | Discrete(2) | [0,500] | CartPole agent |
| Pendulum | (3) | Vector(1) | [-1200,0] | Pendulum agent |
| StarCraft II | (919) | Discrete (165) | -1,1 | StarCraft II bot |
| Connect 4 | (3,6,7) | Discrete(7) | -1,0,1 | First player |
| Tic-Tac-Toe | (3,3,3) | Discrete(9) | -1,0,1 | First player |
| Breakthrough | (8,8) | Discrete(3) | -1,1 | First player |
| DouDizhu | (23, 54) | Discrete(Undertermined) | 0,1 | Landlord |

diagonally. The game could also end in a draw if the grid is full while no player wins. The winning, draw, and losing outcomes of the player correspond to +1, 0, and -1 rewards, respectively.

- **Breakthrough** uses a $8 \times 8$ grid as Figure 12 shows. Two players iteratively move one piece. If the target square is unoccupied, a piece may advance one space either straight forward or diagonally. Only if the square containing the opponent's piece is one step diagonally forward may a piece move into and replace it. The winner is the first player to reach the opponent's home row. A player loses if all of his pieces are taken. A draw is impossible in the Breakthrough game. The winning and losing outcomes of the player correspond to +1 and -1 rewards, respectively.

- **DouDizhu** is a three-party card game as Figure 13 shows. Players begin the game by putting in bids for the "landlord". The other two players will act as "peasants". Being the first player to run out of cards is the game's goal. By removing all of their cards first, the landlord prevails. If one of the peasants removes all of their cards first, the peasants win. The winning and losing outcomes of the landlord correspond to +1 and -1 rewards, respectively.

We also list some basic information (*e.g.,* observation shape, action space, and reward range) about these selected games in Table 2. Besides, we clarify the target agent to explain in Table 2 as well (*i.e.,* except for OpenAI gym and the Atari games, all the other games are multi-agent games, and we intend to one specific party). For the Pong game, we train an agent based on the PPO algorithm to achieve a relatively high reward (*i.e.,* 0.906 in our setting). Regarding two MuJoCo games, we target to explain the runner in You-Shall-Not-Pass and the kicker in Kick-And-Defend. The DRL models of the target agents in the two MuJoCo games are downloaded from `https://github.com/openai/multiagent-competition/tree/master/agent-zoo`. As for OpenAI's CartPole and Pendulum games, we adopt well-trained agents from the stable-baselines3 model zoo: `https://github.com/DLR-RM/rl-baselines3-zoo`. With respect to Connect 4, Breakthrough, and Tic-Tac-Toe, we explain the first player and utilize example codes in OpenSpiel for training RL agents based on the AlphaZero algorithm [12]. The maximum simulation times for the Monte Carlo Tree Search in Connect 4, Breakthrough, and Tic-Tac-Toe are 300, 300, and 20, respectively. Concerning DouDizhu, we target at explaining the landlord and adopt the models trained by DouZero [16], which can be downloaded from `https://github.com/kwai/DouZero`. When it comes to StarCraft II, we use the built-in bot with difficulty A from PySC2: `https://github.com/deepmind/pysc2`.

**Computational Resources.** In our experiment, we use a server with 2 AMD EPYC 7702 64-Core CPU Processors and 4 NVIDIA RTX A6000 GPUs to train and evaluate our method.

## S5.3   Additional Evaluation Results

First, we demonstrate that our design could achieve the monotonicity property by showing the training curves of all games. Second, we provide the agent's performance and `StateMask`'s performance on all games and show the relative value errors across all games. Last, we compare the fidelity scores of our

Table 3: **Results of reward gap in all games.** Each experiment contains 500 runs. We repeat all experiments three times and report the mean and standard deviation of the discounted total rewards in the table below.

| Games | Agent Performance | Relative Errors (%) | Games | Agent Performance | Relative Errors (%) |
|---|---|---|---|---|---|
| Pong | 0.20 (0.00) | 0.89 (0.03) | StarCraft II | 0.11 (0.00) | 0.40 (0.01) |
| You-Shall-Not-Pass | -0.13 (0.01) | 2.58 (0.12) | Connect 4 | 0.91 (0.01) | 2.09 (0.12) |
| Kick-And-Defend | 0.13 (0.01) | 7.78 (0.20) | Tic-Tac-Toe | 0.76 (0.02) | 6.43 (0.08) |
| CartPole | 99.34 (0.00) | 0.00 (0.00) | DouDizhu | 0.40 (0.02) | 14.98 (0.21) |
| Pendulum | -144.16 (10.26) | 5.56 (0.33) | Breakthrough | 0.98 (0.00) | 11.02 (0.16) |

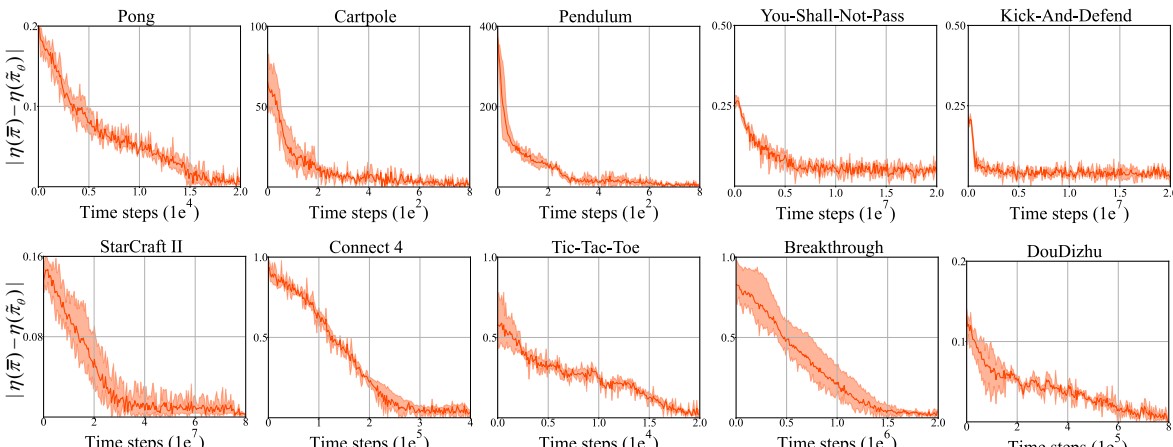

Figure 3: The absolute policy value difference between the masked agent trained by `StateMask` and the target agent. Note that the darker solid lines represent the average absolute policy value difference across 3 random seeds. The lighter shadow indicates the corresponding variations between the maximal and minimal absolute policy value difference.

method with three baseline methods on additional five games, *i.e.,* Kick-And-Defend in MuJoCo games, OpenAI's Cartpole and Pendulum, Breakthrough and Tic-Tac-Toe in Openspiel.

**Monotonicity Property.** Figure 3 shows the trend of $|\eta(\bar{\pi}) - \eta(\tilde{\pi}_\theta)|$ (*i.e.,* the absolute policy value difference) during the training processes of all games. As we can observe, the absolute policy value difference monotonically decreases until it converges when the training step increases. Combined with Section 3, it demonstrates that our design could guarantee the monotonicity property both theoretically and empirically.

**Agent Performance.** To evaluate the agent's performance before and after applying the explanation network, we run 500 tests and record the mean and standard deviation of the discounted total reward in Table 3. We can observe that the agent before and after blinding achieves similar performance when running 500 episode tests. It confirms that our proposed method could achieve the objective in Section 3 (*i.e.,* perturbing the actions in the identified non-important time steps will not impact the expected total reward of the target agent).

**Fidelity.** Section 4 shows the fidelity score comparison between our method and three baseline methods on Pong, You-Shall-Not-Pass, Connect 4, DouDizhu, and StarCraft II. Here, we additionally evaluate the fidelity of our method and three baseline methods on the remaining five games. Figure 4 shows the fidelity score comparison across all four methods. Combined with Figure 3 in Section 4, `StateMask` consistently achieves the highest fidelity score among all games, implying our method's high fidelity.

**Sensitivity.** To show the insensitivity of our proposed method `StateMask`, we vary the hyperparameter $w$ from {0, 1e-5, 1e-4, 1e-3, 1e-2} and train the state mask accordingly. Figure 5 (a) and (b) shows the fidelity scores and mask ratios of our method in Pong and You-Shall-Not-Pass games under different $w$ settings. We can observe that through the hyperparameter $w$ differs, the fidelity scores do not vary too much. It confirms that our method is insensitive to hyperparameter choice. Moreover, adding a sparsity constraint can increase the mask ratio which can help filter out those non-important time steps.

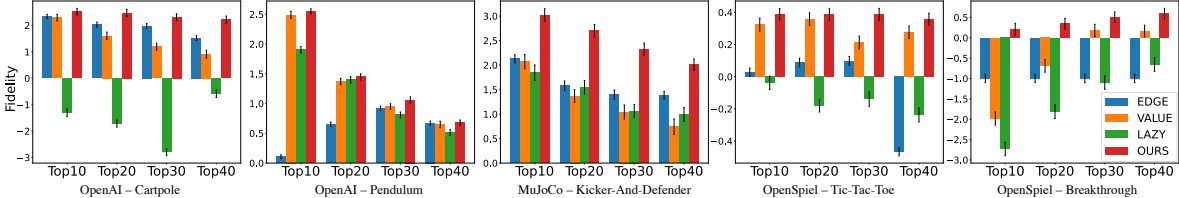

Figure 4: Fidelity scores for interpretations generated by three baseline methods and our proposed explanation method. Note that the definition of the fidelity score can be found in Section 4.2. A higher score implies higher fidelity.

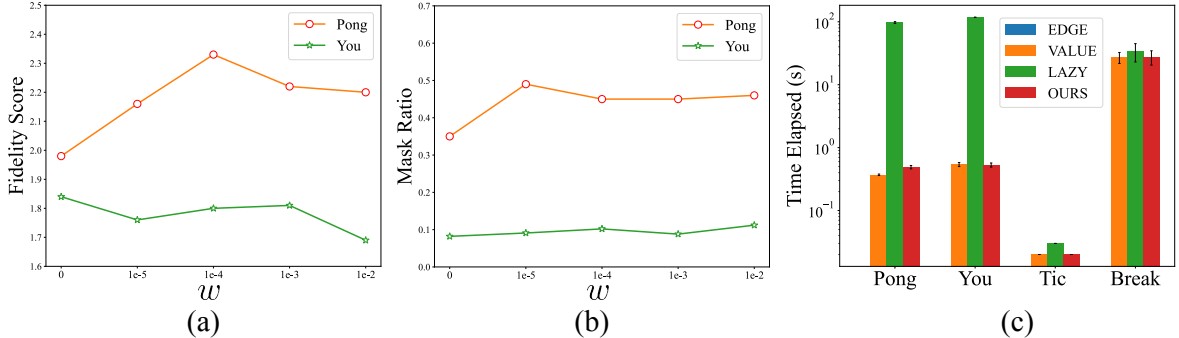

Figure 5: Sensitivity and efficiency results. In the left and middle figures, we show the sensitivity results by varying the hyperparameter $w$ in Pong and You-Shall-Not-Pass games. In the right figure, we report the mean and standard deviation of the explanation time per trajectory in four games: Pong, You-Shall-Not-Pass, Tic-Tac-Toe, and Breakthrough.

**Efficiency.** To demonstrate our method's efficiency, we also record the average explanation time per trajectory in two normal-form games and two extensive-form games. Figure 5 (c) shows the explanation time of each method in four games. Here, EDGE is a global explanation method whose explanation time is negligible while the other three are local explanation methods. Regarding local explanation methods, our proposed method StateMask has similar time efficiency with the Value-max method whereas LazyMDP is much slower. The reason is that the target agents in these four games cannot provide the Q value function directly, and thus we need extra computation time to approximate Q values for deriving explanations in LazyMDP.

# S6   Additional Visualization Results

**You-Shall-Not-Pass.** In Figure 6, we visualize one episode of the You-Shall-Not-Pass game. Our explainer pinpoints the time steps when the red robot escapes the blue robot as the most important ones. The reason is that the DRL agent should be careful not to get knocked down by the opponent at this moment. Otherwise, the agent will lose the balance and fall to the ground, and eventually lose the game. When the two robots are far away from each other or the blue robot is already down on the ground, the states are less important.

**Kick-And-Defend.** We provide one case study for the Kick-And-Defend game in Figure 7. Our explanation method explains that the most important time steps in this game are when the blue robot is shooting. The agent should choose the proper direction and power to kick the ball to the goal at these critical time steps. Otherwise, it will be difficult for the agent to score. When the ball is flying toward the goal, the actions of the DRL agent have no impact on the final result. Therefore, these time steps are less important.

**CartPole.** For the CartPole game, we show an example in Figure 8. When the angle of the pole is large (no matter if it is clockwise or counterclockwise), the cart should accelerate toward the center to correct the angle, otherwise the pole will fall. StateMask treats these time steps as critical ones. When

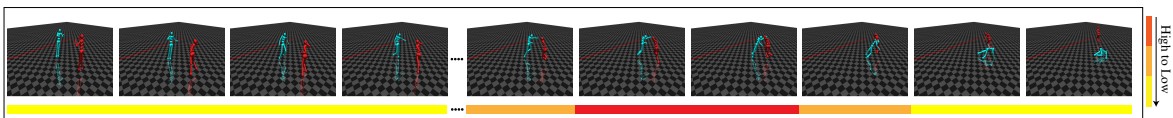

Figure 6: Visualization of the identified critical time steps in You-Shall-Not-Pass game. The DRL agent controls the red humanoid robot and the opponent controls the blue one. While the blue humanoid bot is trying to block the red one from reaching the goal, the red one is trying to get across the red line.

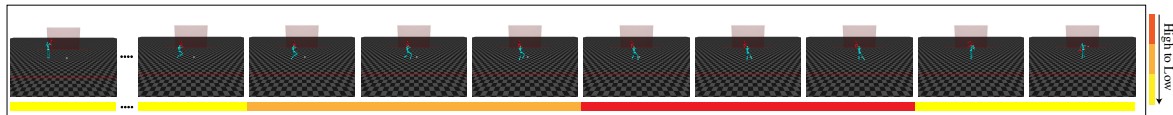

Figure 7: Visualization of the identified critical time steps in Kick-And-Defend game. The DRL agent controls the blue humanoid robot to kick the ball and the opponent controls the red one to defend the goal.

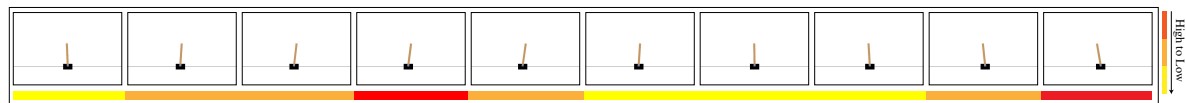

Figure 8: Visualization of the identified critical time steps in Cartpole game. The DRL agent controls the cart to move left or right to keep the pole stand (within a certain angle) as long as possible.

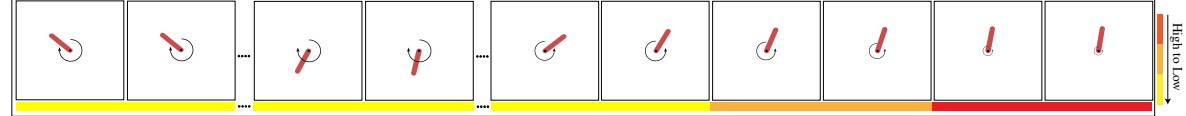

Figure 9: Visualization of the identified critical time steps in Pendulum game. The DRL agent controls the torque applied on the free end of the pendulum to swing it into an upright position, with its center of gravity right above the fixed point.

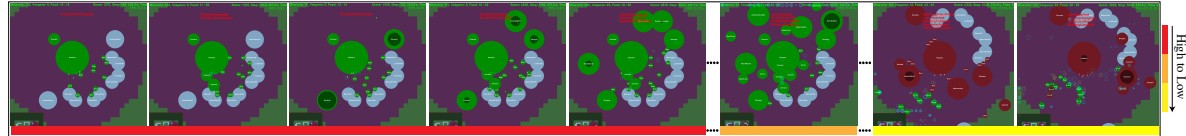

Figure 10: Visualization of the identified critical steps in StarCraft II. The green zerg player is the target agent and the red one is the opponent. The big circles represent the buildings while the small circles represent the units. In the first three steps, the agent collects the minerals and vespene gas. In the next three steps in this episode, the agent builds some structures. In the last two steps, the small circles representing the zerglings and roaches of the target agent are attacking the enemy base.

the angle of the pole is small, the cart needs to stabilize the angle, and `StateMask` treats these time steps as less important ones. This is reasonable since the pole is still within a safe angle range at these time steps.

**Pendulum.** In the Pendulum game, we show one case study in Figure 9. The pendulum starts in a random position and the agent tries to swing it into an upright position. `StateMask` pinpoints the time steps when the pendulum is close to the upright position as the most important ones. The agent should carefully apply the torque at these time steps to make the pendulum upright with zero velocity. When the angle of the pendulum is large, `StateMask` treats these time steps as non-critical ones.

**StarCraft II.** In the StarCraft II game, the DRL agent controls a Zerg player to defeat the opponent Zerg player by collecting resources and building a Zerg army. Figure 10 shows one roach rush game where the DRL agent wins. `StateMask` identifies the first few steps as critical ones, including producing the drones and overlords, as well as building the spawning pool, extractors, and roach warren. These steps are reasonable and will also be deemed as important by human analysis since the successful execution of the build orders in the early stage is one key to winning the game. Also, `StateMask`

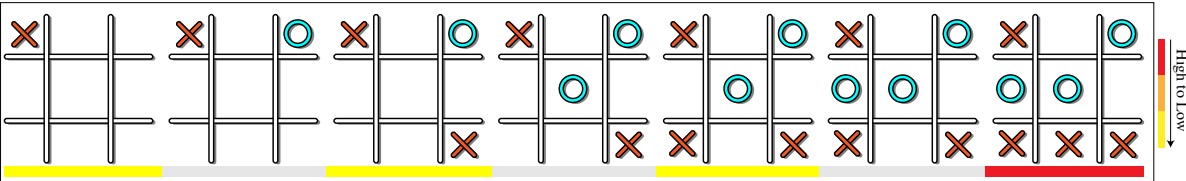

Figure 11: Visualization of the identified critical time steps in the Tic-Tac-Toe game. The DRL agent places the X mark and the opponent places the O alternately in one of the nine spaces in the grid. The states where the O mark is placed are colored in gray.

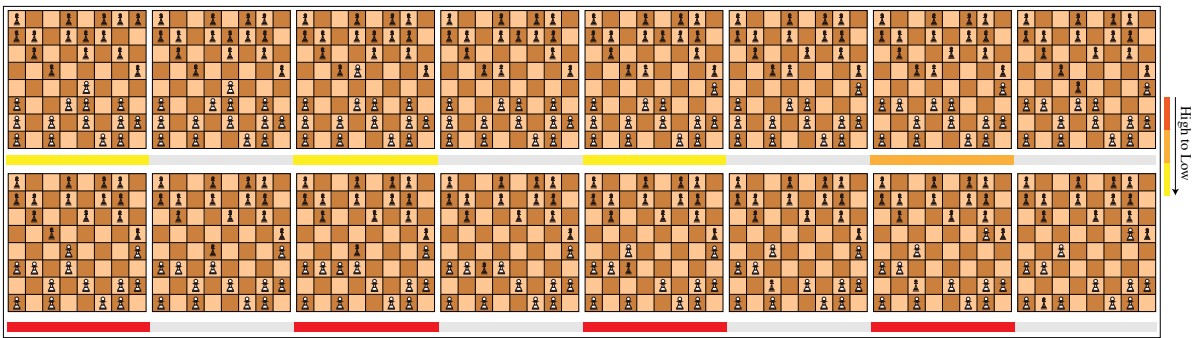

Figure 12: Visualization of the identified critical time steps in Breakthrough game. The DRL agent plays as the black player and the opponent plays as the white one. They play alternatively, with each player moving one piece per turn.

treats the last few steps as non-important steps. This is because the opponent's army has already been crashed by the roach rush, and without the army's protection, it is just a matter of time before the agent destroys the opponent's base to win the game. Because of the automatic attack mechanism in Starcraft II, even if the DRL agent does not take any action, the roaches will destroy all the opponent's buildings eventually.

**Tic-Tac-Toe.** In the Tic-Tac-Toe game, we show one episode to analyze the explanation of our method in Figure 11. Our explainer only selects the last time step as the critical one since this move seals a win for the DRL agent. If the agent chooses the wrong move at this time step, the opponent will win the game.

**Breakthrough.** Finally, we visualize one episode of the Breakthrough game in Figure 12. Our explainer pinpoints the last four time steps in this episode as the most important. By analyzing the strategy of the agent, we could find that the agent sacrifices one piece to make sure another piece can reach the opponent's side in the last four time steps. If the agent does not take such a strategy, the game will last for a longer time and the agent may lose the game eventually.

**DouDizhu.** In DouDizhu, a landlord (our target agent) competes with two cooperative peasants to be the first to run out of cards. Figure 14 shows one critical step identified by `StateMask`. At this step, the landlord chooses to play eight cards, *i.e.,* Quad with Pairs. This move enables the landlord to play most low-rank cards and win the right to play first in the next round (no peasant has larger combinations than this one). If these cards are not played together, it is hard for the landlord to play them separately because the Quad and two Pairs are relatively small and cannot help the landlord to get the right to continue playing first.

We also give the winning trajectory for the DouDizhu game in Figure 13. Our explainer pinpoints two critical time steps in this episode. The first one is when the landlord plays **2**. Since **Red Joker** in hand is the largest solo that can beat any other solo, the landlord can ensure the right to play cards in the next round. The second one is when the landlord plays **Trio with Solo** to empty his's hand. If the landlord plays **Trio** or **Pair** separately, the peasant will have a chance to win the game. Therefore, the two selected time steps are critical to the final result.

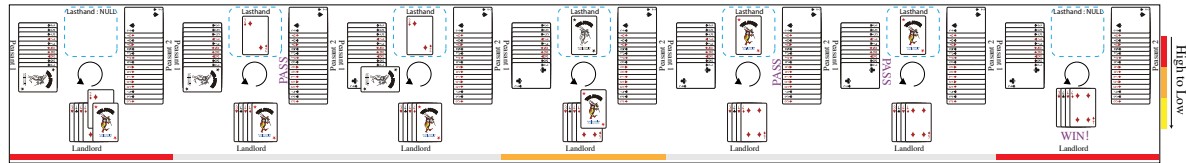

Figure 13: Visualization of the identified critical time steps in the DouDizhu game. The DRL agent plays the role of the landlord and two cooperative opponents play the role of the peasants. The landlord's goal is to win the game by emptying his hand, and vice versa. The states where the peasants take actions are marked with a gray bar.

## S7 Additional Results of Attacks

In this section, we provide the results of launching adversarial attacks under the guidance of explanation when varying the attack ratio between 10% and 30% in Table 4. First, we can observe that when the threshold increases, the victim agent's performance drops. Second, with the same threshold setting, the adversarial attack under the guidance of our explanation method is the most powerful one. These results confirm the conclusion that our proposed explanation method has the highest fidelity.

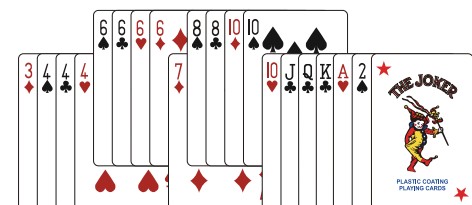

Figure 14: One critical step in the DouDizhu game: The target agent (landlord) plays the Quad with Pairs to ensure that peasants cannot beat this hand.

## S8 Additional Results of Policy Patching

**Advantage of Our Patching Method.** In this experiment, we run the patching method proposed in EDGE [3] on two MuJoCo games. At a high level, this method first replays the losing trajectory until the start of the critical steps and does random explorations at the critical steps in order to find a winning strategy. The actions together with the corresponding critical steps in the winning strategy are recorded. Then, it forms a look-up table with these collected state-action pairs and uses the look-up table to remediate the original policy. We follow the setting in EDGE and run 500 tests to obtain the target agents' winning rate change before and after patching. Table 5 shows the results of EDGE's patching method. Although the agents' performances have some improvements after utilizing the EDGE's patching method, the improvements are overall marginal. Together with the results in Table 1 in Section 5 and Table 6, we can safely conclude the superiority of our proposed patching method over EDGE's patching method after retraining. The reason may be that EDGE uses a hard rule (*i.e.,* look-up table) to patch the agents' policy, and the quality of the look-up table has a great influence on the patched agents' final performance. In contrast, our patching method does not require a fixed look-up table to store the correct actions corresponding to critical states while encouraging the DRL agent to do more exploration at critical time steps and update the neural network parameters accordingly.

**Patching Results on Other Games.** Table 6 provides the agent performance improvement results after retraining under the guidance of our explanation method and the three baselines in Pendulum, Kick-And-Defend, and Tic-Tac-Toe. Using our proposed retraining method under the guidance of explanation, the performance of the reinforcement learning agent generally increases. More importantly, retraining based on our explanation achieves the best performance.

Table 4: Winning rate drop of varying attack ratios in all games. Each experiment contains 500 runs. We repeat all experiments three times and report the average rewards with the corresponding standard deviations in the table below.

| Games | Attack ratio | EDGE | Value-max | LazyMDP | Ours |
|---|---|---|---|---|---|
| Pong | 10% | -63.04 (3.44) | -0.40 (0.40) | -36.17 (1.15) | **-84.90 (1.40)** |
| | 20% | -65.73 (0.81) | -7.06 (8.81) | -57.00 (0.92) | **-85.53 (0.75)** |
| | 30% | -71.07 (1.36) | -70.13 (0.64) | -62.63 (1.63) | **-86.90 (0.76)** |
| You-Shall-Not-Pass | 10% | -2.64 (0.60) | -14.80 (3.20) | -28.87 (1.21) | **-35.93 (1.81)** |
| | 20% | -6.87 (0.61) | -55.13 (4.27) | -50.63 (0.80) | **-61.67 (0.12)** |
| | 30% | -11.68 (1.70) | -61.93 (0.12) | -61.87 (0.23) | **-61.93 (0.12)** |
| Kick-And-Defend | 10% | -10.46 (0.97) | -24.57 (1.86) | -28.33 (1.76) | **-28.37 (1.10)** |
| | 20% | -28.51 (0.72) | -64.23 (1.63) | -60.43 (1.33) | **-68.63 (1.17)** |
| | 30% | -37.25 (1.03) | -74.30 (0.40) | -73.57 (1.53) | **-74.10 (0.80)** |
| CartPole | 10% | -78.98 (4.61) | -4.85 (6.64) | -0.50 (0.51) | **-127.81 (13.82)** |
| | 20% | -153.21 (11.87) | -20.96 (6.21) | -13.92 (6.42) | **-198.06 (24.21)** |
| | 30% | -237.03 (5.30) | -72.99 (7.98) | -56.71 (2.79) | **-332.05 (16.33)** |
| Pendulum | 10% | -58.84 (2.79) | -255.80 (15.92) | -135.23 (1.51) | **-570.77 (1.13)** |
| | 20% | -129.37 (1.86) | -432.15 (43.03) | -241.20 (0.95) | **-707.69 (0.74)** |
| | 30% | -201.39 (3.31) | -563.63 (15.16) | -372.88 (4.21) | **-796.07 (12.07)** |
| StarCraft II | 10% | -70.00 (14.18) | -77.33 (5.69) | -72.00 (13.53) | **-83.00 (7.21)** |
| | 20% | -93.33 (4.73) | -93.00 (3.61) | -93.00 (4.58) | **-94.00 (3.00)** |
| | 30% | -97.00 (0.00) | -97.00 (0.00) | -97.00 (0.00) | **-97.00 (0.00)** |
| Connect 4 | 10% | -33.33 (1.56) | -26.30 (1.44) | -22.57 (1.40) | **-35.37 (3.00)** |
| | 20% | -56.73 (2.30) | -67.57 (1.57) | -53.83 (0.81) | **-71.90 (1.44)** |
| | 30% | -76.23 (1.94) | -82.70 (0.80) | -69.83 (1.17) | **-85.50 (0.72)** |
| Tic-Tac-Toe | 10% | -0.30 (0.23) | -8.07 (0.35) | -4.40 (0.25) | **-17.24 (0.40)** |
| | 20% | -6.72 (0.33) | -18.20 (0.17) | -7.60 (0.12) | **-29.90 (0.45)** |
| | 30% | -19.50 (0.36) | -25.12 (0.13) | -13.90 (0.41) | **-42.85 (0.17)** |
| Breakthrough | 10% | -6.53 (0.50) | -18.00 (1.00) | -18.90 (0.36) | **-88.73 (1.62)** |
| | 20% | -13.67 (0.83) | -30.93 (1.01) | -26.87 (1.03) | **-95.87 (1.21)** |
| | 30% | -26.33 (1.53) | -42.67 (2.89) | -36.00 (2.65) | **-98.33 (1.15)** |
| DouDizhu | 10% | -0.25 (1.71) | -18.60 (0.31) | -25.20 (0.76) | **-29.94 (0.64)** |
| | 20% | -10.45 (1.68) | -24.40 (0.83) | -32.27 (0.69) | **-32.87 (0.20)** |
| | 30% | -11.29 (1.40) | -28.07 (1.22) | -32.67 (0.20) | **-34.54 (0.31)** |

# S9 Evaluation on Explaining Sub-optimal Agents

As mentioned in Section 4.3, to verify the effectiveness of `StateMask` in terms of explaining sub-optimal agents, we conducted additional experiments on sub-optimal agents in Pong, Kick-And-Defend, Connect 4, and Breakthrough games. These games include both normal-form games (*i.e.,* Pong and Kick-And-Defend) and extensive-form games (*i.e.,* Connect and Breakthrough). Regarding the selection of sub-optimal agents in these four games, we utilized policies from the halfway stage of training as sub-optimal agents in Pong, Connect 4, and Breakthrough. In the Kick-And-Defend game, we follow the setting in Supplement S5.2 and explain the defender. Note that the original zoo model (`https://github.com/openai/multiagent-competition/tree/master/agent-zoo`) offered three different policies for the kicker. We held the defender constant and had each kicker play against the defender separately, recording their corresponding winning rates. The kicker that displayed the lowest winning rate was chosen as the sub-optimal agent.

## S9.1 Fidelity

In this experiment, we compare the fidelity scores of our method and baseline methods across the four selected games. As depicted in Figure 15, our method still achieves the highest fidelity score among the four games even though the target agent is sub-optimal. The reason is that `StateMask` is designed to preserve the original agent's performance before and after masking the target agent at

Table 5: Results of EDGE's policy patching method with different explanation methods. Each experiment contains 500 runs. We repeat all experiments three times and report the average reward increase with the corresponding standard deviation in the table below.

| Application | Games | EDGE | Value-max | LazyMDP | Ours |
|---|---|---|---|---|---|
| Target agent's winning rate improvement after patching | Kick-And-Defend | +0.00 (0.00) | +0.00 (0.00) | +0.00 (0.00) | +0.00 (0.00) |
| | You-Shall-Not-Pass | +0.00 (0.00) | +0.75 (0.04) | +0.25 (0.01) | +0.25 (0.02) |

Table 6: Additional results of policy patching. Each experiment contains 500 runs. We repeat all experiments three times and report the average reward increase with the corresponding standard deviation in the table below.

| Application | Games | EDGE | Value-max | LazyMDP | Ours |
|---|---|---|---|---|---|
| Target agent's winning rate improvement after patching | Kick-And-Defend | +1.03 (0.12) | +0.70 (0.20) | +0.97 (0.06) | **+1.37 (0.12)** |
| | Pendulum | +0.00 (0.00) | +0.29 (0.03) | +2.65 (0.05) | **+4.30 (0.40)** |
| | Tic-Tac-Toe | +0.03 (0.06) | +0.23 (0.06) | +0.20 (0.10) | **+ 2.47 (0.12)** |

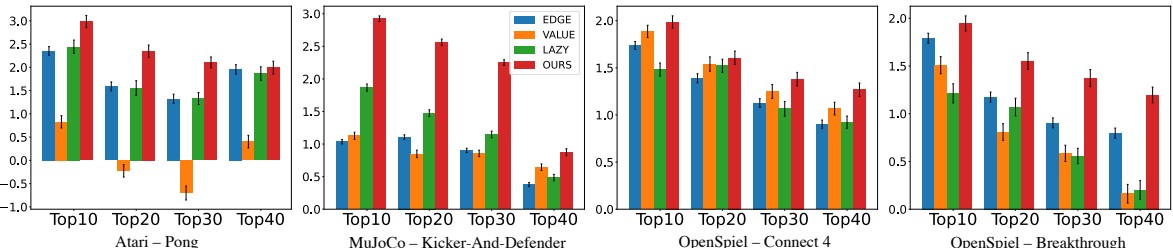

Figure 15: Fidelity scores for interpretations generated by three baseline methods and our proposed explanation method for four sub-optimal agents.

certain time steps. In instances where the target agent is sub-optimal, the important time steps are those that lead the target agent to win or lose (*i.e.,* with a higher chance) in a single episode. Altering the actions in these time steps may cause a significant variation in performance. Therefore, StateMask has the potential to recognize and distinguish these critical time steps by learning how to unmask them accurately.

## S9.2  Visualizing the Explanation for Sub-optimal Agent Behaviors

Following Section 5, we also visualize the important time steps identified by StateMask in the selected four games to demonstrate StateMask could generate human-understandable explanations even for sub-optimal agents.

**Pong.** Regarding the Pong game as Figure 16 shows, our method indicates that the most crucial time step is when the ball is flying towards the yellow paddle. Nevertheless, the yellow paddle moves upward too much when trying to catch the ball. If the yellow paddle takes a different action, *e.g.,* moves downward, the game outcome could be flipped.

**Kick-And-Defend.** In the Kick-And-Defend game, as our explanation method demonstrates in Figure 17, the most critical time step is identified when the kicker is in contact with the ball. As shown in the figure, the kicker's positioning and approach to the ball lead to a sub-optimal angle for the shot. Consequently, this results in the ball being intercepted easily by the defender, who is well-positioned to block the trajectory. If the kicker were to adjust their angle or approach, the outcome of the game could potentially change, as the ball might bypass the defender, increasing the likelihood of scoring a goal.

**Connect4.** In the Connect 4 game as Figure 18 shows, StateMask pinpoints the penultimate step as the most crucial step of the losing trajectory. Once the black player determines to drop his token into the seventh column in the second-to-the-last step, we could observe that no matter which column the black player chooses in the last step, the game outcome will always be a failure. However, if the black player chooses the second column in the second-to-the-last step, he would have a chance to win,

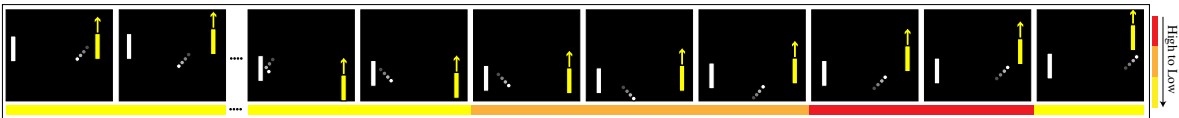

Figure 16: Visualization of the identified critical time steps in the Pong game. The yellow paddle is controlled by the agent we aim to explain.

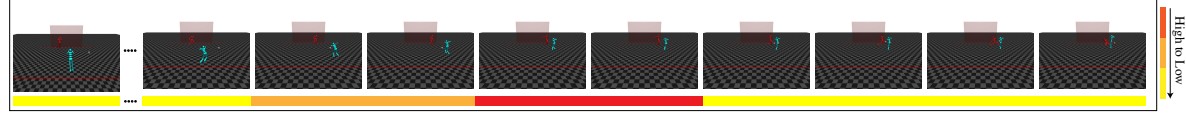

Figure 17: Visualization of the identified critical time steps in the Kick-And-Defend game. The kicker(blue agent) is the agent we target to explain.

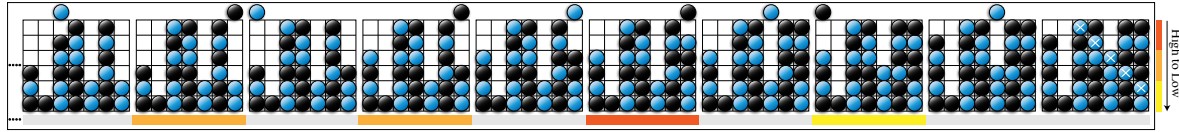

Figure 18: Visualization of the identified critical time steps in Connect 4 game. The black player is the sub-optimal agent we aim to explain.

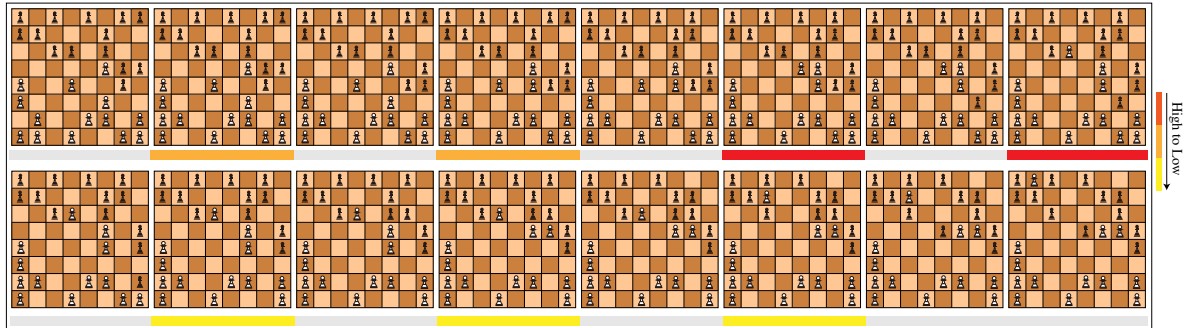

Figure 19: Visualization of the identified critical time steps in the Breakthrough game. The sub-optimal agent plays as the black player and the opponent plays as the white one.

*i.e.,* , connecting four tokens between the first column and the fourth column either horizontally or diagonally. Therefore, the second-to-the-last step is the most crucial step in this losing trajectory.

**Breakthrough.** Regarding the Breakthrough game as Figure 19 shows, our method identifies the fifth-to-the-last and the fourth-to-the-last step as the most critical ones. Note that only if the square containing the opponent's piece is one step diagonally forward may a piece move into and replace it. Apparently, in the fifth-to-the-last step, the black player fails to replace the left white piece in the fourth row, directly causing the game to lose. Additionally, in the fourth-to-the-last step, the black player's action makes him lose the most advanced piece. The identified two steps are thus the key steps for the black player to lose the game.

## S9.3    Patching Results of Sub-optimal Agents

In this experiment, we follow the retraining method proposed in Section 5 and patch the sub-optimal agents in the selected four games. Table 7 shows the results of the sub-optimal agent performance improvement under the guidance of both `StateMask` and the baseline methods. After retraining, the agents' performance generally has a large improvement due to these agents being far from optimal. More importantly, retraining based on our explanation achieves the best performance, which indicates the superiority of `StateMask`.

Table 7: Patching results of sub-optimal agents in four games. Each experiment contains 500 runs. We repeat all experiments three times and report the average reward increase with the corresponding standard deviation in the table below.

| Application | Games | EDGE | Value-max | LazyMDP | Ours |
|---|---|---|---|---|---|
| Target agent's winning rate improvement after patching | Pong | +5.80 (0.12) | +5.40 (0.03) | +12.00 (1.24) | **+13.60 (1.22)** |
| | Kick-And-Defend | +7.42(0.40) | +8.43(0.24) | +4.78(0.12) | **+10.26(0.62)** |
| | Connect 4 | +24.80 (1.68) | +35.80 (1.84) | +36.20 (2.20) | **+37.60(1.66)** |
| | Breakthrough | +68.40 (3.68) | +65.80 (2.77) | + 68.80 (3.26) | **+69.00 (3.07)** |

Table 8: Results of relative errors in four selected games. Each experiment contains 500 runs. We repeat all experiments three times and report the mean and standard deviation of the discounted total rewards in the table below.

| Games | | Pong | Kick-And-Defend | Connect 4 | Breakthrough |
|---|---|---|---|---|---|
| Sub-optimal agent performance | | 0.18 (0.00) | 0.09 (0.00) | 0.52 (0.01) | -0.39 (0.01) |
| Relative error | StateMask | 2.57 (0.03) | 5.34 (0.54) | 3.05 (0.34) | 12.89 (1.86) |
| | PPOMask | 3.86 (0.06) | 7.53 (0.65) | 5.77 (0.42) | 28.57 (2.44) |

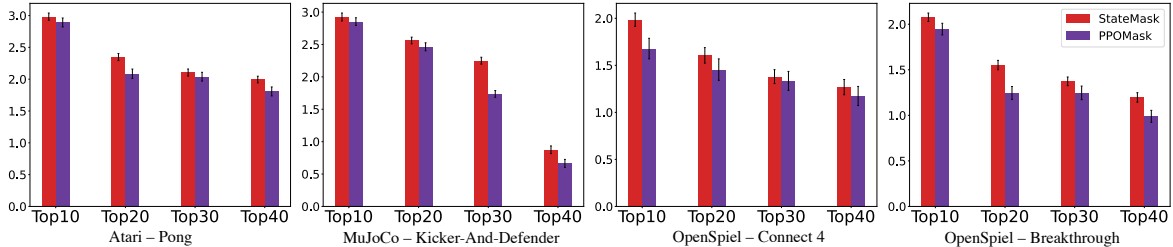

Figure 20: Fidelity comparisons between StateMask and PPOMask across four selected games when the agent is sub-optimal.

# S10 Evaluation on An Alternative Design

As mentioned in Section 4.3, we examine an alternative design of StateMask named as PPOMask that trains a mask network by maximizing the masked agent's expected total reward (*i.e.*, $J(\theta) = \max_\theta \eta(\tilde{\pi}_\theta)$). This is done by directly using the PPO algorithm to train the mask network. To avoid the trivial solution, same as Section 3.2, we also add an additional loss item $L_t^{\mathrm{MASK}}$ to the PPO loss. We evaluate PPOMask in Pong, Kick-And-Defend, Connect 4, and Breakthrough game.

**Relative Error.** Here, we report the relative errors of PPOMask in comparison to StateMask across these chosen four games. As depicted in Table 8, PPOMask generates a higher relative error than StateMask when the target agent is sub-optimal, due to its inability to maintain the target agent's performance (*i.e.*, PPOMask trains a masked agent that performs better than the original agent).

**Fidelity.** We also test the fidelity of PPOMask and compare it with StateMask in these four games. According to Figure 20, PPOMask exhibits poor fidelity on sub-optimal agents. This can be attributed to the objective function of PPOMask, which is to maximize the expected total reward of a mixed policy. Consequently, PPOMask might disregard the critical states that contribute to the failure of the original agent and instead take random actions (*i.e.*, mask operation) during those steps in order to secure a win under the perturbed policy. However, as shown in S9, our method is able to identify the critical states that contribute to the failure of the sub-optimal agent and thus has higher fidelity.

# S11 Potential Social Impact

Our work focuses on improving the trustworthiness of Deep Reinforcement Learning systems, which are increasingly being used in a wide range of applications, from gaming [5] to robotics [8] to healthcare [15].

One of the main challenges of DRL is the black-box nature of the models, which makes it difficult to understand how they arrive at their decisions. This lack of transparency can create concerns about the reliability and safety of DRL systems.

To address this challenge, our work aims to automatically generate explanations for DRL systems, without requiring human supervision. By generating explanations automatically, we reduce the cost and effort required to provide explanations, making them more accessible and scalable for various applications. Moreover, it avoids potential biases that may arise from human-generated explanations, as these may be influenced by subjective factors or an incomplete understanding of the system.

Our approach involves developing models that can analyze the behavior of DRL systems and generate explanations based on that analysis. These explanations provide insights into how the DRL system works, what factors influence its decisions, and how it can be improved. By providing this information, our work contributes to enhancing the transparency and interpretability of DRL systems, which can help users to evaluate the reliability of these systems.

When it comes to the potential negative societal impacts of our work, the use of `StateMask` for enhancing RL agents may lead to the magnification of biased and unfair RL agents. Nevertheless, our method also has the capability of understanding the decision-making process of biased and unfair RL agents, which enables the implementation of strategies to diminish their harmful effects. Therefore, `StateMask` is crucial for understanding and improving RL agents.

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
