# OpenReview forum: "StateMask: Explaining Deep Reinforcement Learning through State Mask"
_NeurIPS.cc/2023/Conference — NeurIPS 2023 poster_

### Official Review · Reviewer_AA7o · 2023-07-02

**Soundness:** 3 good
**Presentation:** 3 good
**Contribution:** 3 good
**Rating:** 6
**Confidence:** 4

**Summary:**

The paper presents an interesting model (so called statemask) to identify critical states for an agent's final reward. The goal of statemask is to find the non-important time steps and randomize their actions without changing the expected total reward of the target agent. A PPO based algorithm is leveraged to formally generate the model. Several numerical examples are shown to demonstrate the merits of the proposed model.

**Strengths:**

The paper presents an interesting model (so called statemask) to identify critical states for an agent's final reward. The goal of statemask is to find the non-important time steps and randomize their actions without changing the expected total reward of the target agent. Several numerical examples are shown to demonstrate the merits of the proposed model.

**Weaknesses:**

The paper seems claim the method is suitable for all decision making processes. However, for the type of shortest path finding problems, it is questionable that the method will be effective. In fact, the critical elements are not states, but the critical paths. So, it would be interesting to know statemask fits to what kinds of processes which is missing in the current version.

**Questions:**

The optimization of (2) makes sense to provide a mask network. However, it is interesting to know if the state space is huge, how to find those critical states. For example, how the critical states in Figure 4 and Figure 5 are identified?

---

> ### Author Rebuttal · Authors · 2023-08-09
>
> We thank Reviewer AA7o for the constructive and insightful comments. Please see our response to each of your questions below.
>
> **1. The paper seems to claim the method is suitable for all decision-making processes. However, for the type of shortest path finding problems, it is questionable whether the method will be effective. In fact, the critical elements are not states, but the critical paths. So, it would be interesting to know statemask fits to what kinds of processes which is missing in the current version.**
>
> We do not claim that our method is suitable for all decision-making processes. Our method may not be suited to explain the agent with poor performance. In this case, the mask net may not differentiate the important and unimportant states since replacing the current action with random action or not will have little influence on the final performance.
>
> While our method is to identify the states most critical to the expected total reward, it is also implicitly associated with the related action at these states. Throughout treating the target agent with a fixed policy as a part of the environment as Figure 2 shows, StateMask actually takes both the state and the corresponding action of the target agent into account. Therefore, StateMask indirectly figures out the importance of the state-action pair. Back up to the shortest path finding problem, by identifying these crucial state-action pairs, we can reconstruct the corresponding path. Specifically, we apply our method in a shortest path finding environment--MiniGrid-Empty-6x6-v0 [1]. As is shown in Figure 2 of the attached PDF,  the goal of the agent is to reach the green goal square using as few steps as possible. When the agent is optimal as trajectory 1 shows, our method figures out that all steps in the path are important. When the agent is sub-optimal as trajectories 2 and 3 show, our method highlights the path starting from the last sub-optimal action as the most critical one since in the pinpointed critical path, the agent must take actions to reach the green goal square with the fewest steps possible to keep the same reward. These insights underline the adaptability of StateMask to the shortest path problem and highlight its ability to capture critical steps in diverse decision-making contexts.
>
> [1] Minimalistic Gridworld Environment (MiniGrid). https://github.com/maximecb/gym-minigrid.
>
> **2. The optimization of (2) makes sense to provide a mask network. However, it is interesting to know if the state space is huge, how to find those critical states. For example, how the critical states in Figure 4 and Figure 5 are identified?**
>
> Our method is independent of the size of the state space. As stated in Sec. 3, we measure the significance of states based on the probability of StateMask output being 0, i.e., the probability of keeping the target agent’s action. Take Figure 4 and Figure 5 as an example, for a single trajectory, we calculate the importance score of each time step using StateMask and then rank the corresponding importance scores from top to bottom to identify these critical states.
>
> In addition, as depicted in Figure 4, we successfully apply our StateMask approach to the Pong game, which has a large state space ($210\times160\times3$). Despite the complexity of the state space, our method accurately predicts the importance of states with high fidelity as Figure 3 in the main text indicates. Moreover, in the paper, we demonstrate the effectiveness of our approach in the MuJoCo game, which has an infinite number of states due to its continuous state space. This result further validates that our approach remains unaffected by the size or complexity of the state space.

---

> > ### Comment · Reviewer_AA7o · 2023-08-12
> >
> > Thanks the authors for additional numerical experiments and detailed explanations for my concerns. Now the contributions of the paper are clear.

---

> > > ### Author Response · Authors · 2023-08-13
> > > **Response to Reviewer AA7o**
> > >
> > > We sincerely appreciate your positive feedback on the additional experiments and detailed explanations we provided. It's gratifying to know that our efforts have contributed to clarifying the contributions of the paper. We will include the additional experiment results and the detailed explanations in the next version of our manuscript.

---

### Official Review · Reviewer_qciU · 2023-07-03

**Soundness:** 2 fair
**Presentation:** 2 fair
**Contribution:** 2 fair
**Rating:** 6
**Confidence:** 4

**Summary:**

This paper focuses on providing an explanation for deep reinforcement learning agents by identifying the important time steps within an episode. The authors propose a module called StateMask, which replaces the original agent's policy with random actions in specific time steps. By preserving the overall episode returns, only non-important time steps are randomized.

**Strengths:**

One notable strength of this paper is the intriguing concept of masking actions, which allows for identifying important time steps without altering the learned agent or its learning process.

**Weaknesses:**

1. The objective Eqn. 2 is problematic. The optimal solution of Eqn. 2 is $\pi=\bar{\pi}$, i.e. $\tilde{\pi}(a_t^e=0|s_t)=1$, where the integration policy $\pi$ degenerate to the target policy $\bar{\pi}$. If StateMask $\tilde{\pi}$ becomes a constant policy, it fails to identify any important time steps.

1. Even if Eqn. 2 cannot be optimized to zero but instead reduced to a small value, StateMask $\tilde{\pi}$ would tend  to predict $a_t^e=0$ in most states but $a_t^e=1$ in a few specific states. Since  $a_t^e=1$ indicates a non-important state, this method can only identify a limited number of non-important states and fails to capture important time steps. Identifying non-important states is not consistent to the major motivation of this paper.

1. The motivation behind using the absolute error in Eqn. 2 and its surrogate objective in Section 3.2 is not clear. A more straightforward approach for regression tasks would be to minimize the squared error (MSE / L2 loss) rather than the L1 loss, as the L2 loss is differentiable everywhere.

1. The true optimization challenge is that the optimization problem of StateMask is another reinforcement learning problem. The authors propose a PPO-like surrogate objective which maximize the return under the constrain of minimizing the difference to target policy $\bar{\pi}$. However, it seems not consistent to the major objective Eqn 2.

**Questions:**

1. The authors should clarify the motivation for proposing the surrogate objective.

1. In line 61, authors claim that they provide theoretical analysis for StateMask but it is missing.

1. What is the definition of $\tilde{\pi}_{\theta}$ in line 195? In the previous sections, $\tilde{\pi}$ denotes the policy of MaskState instead of the integration policy $\pi$.

1. What reward function is used for the advantage function in line 201?

1. The sign in Eqn 8 should be reversed.

1. The metric "fidelity" should be well-defined in the text.

1. The authors should use more metrics to evaluate their method, because StateMask directly optimizes "fidelity" which is not fair to other methods.

**Limitations:**

The authors should refine the objective Eqn 2. Besides, the writing skill needs to be improved.

---

> ### Author Rebuttal · Authors · 2023-08-09
>
> We thank Reviewer qciU for the constructive and insightful comments. Please see our response to each of your questions below.
>
> **1. Questions about Eqn. (2)**
>
> **Q(1):** The optimal solution of Eqn. (2) is $\pi=\bar{\pi}$, failing to identify any important time steps.
>
> Please note that our goal is to find a non-trivial solution (i.e., mask some specific states) to Eqn. (2). The challenges of finding such a solution are explained in line 162-176, and our approach is designed explicitly to seek these “non-trivial” solutions. To achieve this, we introduce a mask ratio constraint (i.e., enforce the mask ratio is larger than a threshold $c$) and propose Eqn. (5) as our objective function. A comprehensive analysis of Eqn. (5) is available in Supp. S2.
>
> **Q(2):** StateMask's limited ability to identify important time steps is not consistent with the paper's major motivation.
>
> Instead of directly using the StateMask output $a_t^e$ to identify important time steps, we evaluate the significance of states based on the probability of mask net output being 0 as stated in Sec. 3.  We calculate an importance score for each time step in a trajectory based on the probabilities given by the mask network's output. These scores are ranked in descending order, enabling us to identify the states that contribute the most to the final reward in the fidelity test, regardless of whether the mask network's output is frequently 0. Consequently, even in situations where the ratio of time steps being masked is low (e.g., <10% in MuJoCo games), the ranking of importance scores enables us to identify the critical time steps effectively (see Figure 3).
>
> **Q(3):** Why use $L_1$ loss instead of $L_2$ loss in Eqn. (2)?
>
> Recall our goal is to minimize the performance difference between two policies $\pi$ and $\bar{\pi}$ , which requires a $L_{p}$ loss to measure the performance gap. The above optimization problem will involve the gradient of $\eta(\pi)$ w.r.t $\theta$. However, a common solution using the standard policy gradient method [1] to optimize $\theta$ cannot guarantee the monotonic decrease of the $L_{p}$. The key challenge is to derive a surrogate objective that enables the monotonically decrease during training. Indeed, whether we use $L_1$ or $L_2$ loss is inconsequential to our algorithm, and Theorem 1 remains valid when optimizing with $L_2$ loss. The objective function, Eqn. (6), will be identical regardless of using $L_1$ or $L_2$ loss. Therefore, without loss of generality, we use $L_1$ loss in Eqn. (2) to introduce the challenge and motivate our design.
>
> [1] Simple statistical gradient-following algorithms for connectionist reinforcement learning. Machine learning,1992.
>
> **2. Motivation of the surrogate objective Eqn. (6) and consistency with Eqn. (2).**
>
> The motivation behind the surrogate objective Eqn. (6) is to address the challenge of obtaining a non-trivial solution for Eqn. (2) with a monotonic decrease guarantee as naively optimizing Eqn. (2) may not guarantee convergence. To solve Eqn. (6), we use a prime-dual method to update $\theta$ and $\lambda$ by solving Eqn. (7) and Eqn. (8). As for Eqn. (7), although it resembles the PPO, it introduces a Lagrange multiplier $\lambda$. It is worth noting that when $\lambda$ is larger than 1, we are doing something like minimizing the return. In fact, $\lambda$ serves as a role to control the performance difference between $\bar{\pi}$ and $\tilde{\pi}$. By updating both $\theta$ and $\lambda$ iteratively, we could solve Eqn. (6) and thereby monotonically decrease the major objective Eqn. (2).
>
> **3.  Lack of theoretical analysis.**
>
> We apologize for any confusion caused. We delve into theoretical analysis in detail in Sec. 3.2. Specifically, we describe the construction of the surrogate objective, which is theoretically derived to guarantee a monotonic decrease in Eqn. (2), in accordance with Lemma 1 and Theorem 1. Furthermore, we provide a comprehensive theoretical analysis in Supp. S2, explaining why the optimization of Eqn. (5) leads to a monotonic decrease in Eqn. (2).
>
> **4. Definition of $\tilde{\pi}_\theta$ in line 195.**
>
> $\tilde{\pi}_\theta$ represents a parameterized policy of StateMask. It's essential to note that the parameterization is applied to $\tilde{\pi}$ and not the integral policy, $\pi$. Please see line 190-193 for more details.
>
> **5. Reward function used for the advantage function in line 201.**
>
> The reward function of the state mask is the same as the environment’s reward provided to the target agent.
>
> **6. Reverse the sign in Eqn. (8).**
>
> Please note that we have negated the sign of the formula in parentheses associated with $\lambda$ when deriving from Eqn. (6)  to Eqn. (8).
>
> **7. The metric "fidelity" should be well-defined in the text.**
>
> We have included a detailed description of the metric "fidelity" in  Supp. S5.1.
>
> **8. Need more metrics to evaluate their method, because StateMask directly optimizes "fidelity" which is not fair to other methods.**
>
> Our optimization objective aims to minimize the performance gap between the target agent and StateMask at the policy level, specifically, by reducing the expected total reward difference between their respective policies. In contrast, "fidelity" is defined at the trajectory level, where we assess if altering a consecutive series of actions within a fixed trajectory results in a significant change in the reward. Hence, StateMask does not directly optimize for "fidelity."
>
> It is worth noting that, to the best of our knowledge, the fidelity score proposed in [2] is the only quantitative metric available in our setting. Additionally, we present qualitative evaluation results in  Supp. S11, where our method surpasses baseline approaches in facilitating the user's understanding of a DRL agent's policy.
>
> [2] Edge: Explaining deep reinforcement learning policies. In Proc. of NeurIPS, 2021.

---

> > ### Author Response · Authors · 2023-08-18
> > **Follow up with Reviewer qciU**
> >
> > We wish to extend our heartfelt appreciation to Reviewer qciU once more, for generously offering us your remarkably insightful comments. As we approach the conclusion of the discussion phase, we take this opportunity to inquire if there exist any queries you would like to share concerning our response. We are pleased to address any further concerns you may have.
> >
> > Furthermore, if our response has satisfactorily addressed your concerns, we kindly ask if you can reconsider your score. Thank you for your time and attention.

---

> > ### Comment · Reviewer_qciU · 2023-08-20
> > **Reply to authors**
> >
> > I appreciate the response from reviewers. It enhances my understand the challenges of non-trivial solution and the monotonicity.
> >
> > 1. One concern I'd like to address is that this method does not propose a surrogate objective but changes the objective. The analysis in Theorem 1 and Lemma 2 lead to infer a constraint
> > $\eta(\tilde{\pi}_{old})\leq\eta(\tilde{\pi})\leq \eta(\bar{\pi})$.
> > Eqn.5 can be regarded as a surrogate objective of
> >
> > $$
> > max_\theta \ |\eta(\tilde{\pi}) - \eta(\bar{\pi})|, s.t.  \ \eta(\tilde{\pi})\leq \eta(\bar{\pi})
> > $$
> > which is equivalent to
> > $$
> > max_\theta \ \eta(\bar{\pi}) - \eta(\tilde{\pi}), s.t.  \ \eta(\tilde{\pi})\leq \eta(\bar{\pi})
> > $$
> >
> > This is different than the original problem (Eqn.2). This problem may converge to a different optimal point due to the absence of absolute value. Besides, this problem has no challenge of monotonicity guarantee stated in line 162-176.
> >
> > 2. The presentation of line 162-176 has to be refined. 1) the challenge is not differentiable or absolute value, as mentioned in Q(3). 2) explain why monotonicity guarantee is required for a regression objective in RL.
> > 1. Why $\omega$ in Eqn(7) is a hyper-parameter but $\lambda$ is Eqn(6) is a parameter to be optimized?
> >
> > 1. I cannot locate the code for optimizing Eqn(8) in "Pong" and in folder "perfect_game". Please help to pinpoint the specific lines.
> >
> >
> > I will consider to raise my score if authors further address these questions.

---

> > > ### Author Response · Authors · 2023-08-21
> > > **Reply to Reviewer qciU (Part 1)**
> > >
> > > We appreciate your thoughtful feedback and your careful consideration of our work. Please see our response to each of your questions below.
> > >
> > >
> > > Regarding your first question, it is worth noting that Lemma 2 infers a constraint $\eta(\tilde{\pi}\_{\theta_{old}}) \leq \eta(\tilde{\pi}\_{\theta}) \leq 2\eta(\bar{\pi})-\eta(\tilde{\pi}\_{\theta_{old}})$, which is different from what you derived.  Eqn. (5) can be regarded as a surrogate objective of $\max\_\theta  \eta(\tilde{\pi}\_{\theta}),  s.t. \eta(\tilde{\pi}\_{\theta}) \leq 2\eta(\bar{\pi})-\eta(\tilde{\pi}\_{\theta_{old}})$ when solved by TRPO. We have shown that a policy $\tilde{\pi}_\theta$ solved from the optimization objective Eqn. (5) satisfies Lemma 2 and thus enables the desired monotonicity of Eqn. (2) as shown in Supp. S2. Therefore, Eqn. (5) is consistent with the original problem Eqn. (2).
> > >
> > >
> > > Regarding your second question, we would kindly like to commence by offering some clarification. It is important to note that we are not asserting that the challenge pertains to the differentiability of an $L_1$ function. For the $L_1$ loss, it is non-differentiable only at the point 0, which coincides with the optimal solution. The depiction provided in lines 163-166 (specifically, the comparison between $\eta(\pi)$ and $\eta(\bar{\pi})$) mirrors the process of taking the gradient of the $L_1$ loss at a non-zero point. We utilize the $L_1$ loss as an illustrative example to highlight that straightforwardly optimizing Eqn. (2) does not ensure a monotonic decrease in Eqn. (2).
> > >
> > > Regarding the necessity of ensuring a monotonically guaranteed outcome, if we directly solve optimization of Eqn. (2), as expounded in our explanation in Q(3), it will entail calculating the gradient of $\eta(\pi\_\theta)$ with respect to $\theta$. Our only recourse to estimating $\nabla\_\theta  \eta(\pi_\theta)$ is through the vanilla policy gradient method $\nabla\_\theta \eta(\pi_\theta) =\nabla_\theta \sum_{t=0}^{T-1} \log \pi_\theta (a_t \mid s_t) \hat{A}_t$ [1] where $\hat{A}_t$ is the estimate of the advantage function at time $t$. Nevertheless, this approach is widely regarded as ill-suited for most problems due to its elevated sample complexity [2,3]. Furthermore, selecting an appropriate step size (i.e., learning rate) that is effective throughout the optimization process for the vanilla policy gradient method becomes a more challenging task. To address these issues, drawing inspiration from TRPO, we have formulated an alternative surrogate objective that facilitates the monotonic decrease of Eqn. (2). We will include a discussion about the necessity of monotonicity in our next version. Thank you for your invaluable suggestion.
> > >
> > > [1] Simple statistical gradient-following algorithms for connectionist reinforcement learning. Machine learning,1992.
> > >
> > > [2] Optimizing expectations: From deep reinforcement learning to stochastic computation graphs. Diss. UC Berkeley, 2016.
> > >
> > > [3] Trust region policy optimization. In Proc. of ICML, 2015.

---

> > > > ### Comment · Reviewer_qciU · 2023-08-21
> > > > **Reply to Authors**
> > > >
> > > > Thanks for reviewer's response. My concerns are addressed so I raise score to 6.

---

> > > > > ### Author Response · Authors · 2023-08-21
> > > > >
> > > > > Thank the reviewer for the positive response! We will add the rebuttal changes to our next version.

---

> > > ### Author Response · Authors · 2023-08-21
> > > **Reply to Reviewer qciU (Part 2)**
> > >
> > > Regarding your third question, we appreciate your observation that $w$ in Eqn. (7) is a hyper-parameter but  $\lambda$ is Eqn. (6) is a parameter to be optimized. Eqn. (5) transforms the original problem Eqn. (2) to a **constrained** reinforcement learning problem (i.e., by constraining the $\eta(\tilde{\pi})$ with an upper bound). To solve Eqn. (5), we transform the performance-bound constraint to a Lagrangian form by following the common strategy in the constrained RL area [4,5].  As such, $\lambda$ is introduced in Eqn. (6) as a Lagrangian multiplier to optimize the Eqn. (5).  As for the sparsity constraint in Eqn. (5), we can also transform the sparsity constraint to the Lagrangian. However, an increasing number of Lagrange multipliers will increase the complexity of optimization [4, 6]. Since the sparsity constraint is essential for us to find a non-trivial solution to Eqn. (2), we are especially interested in accurately controlling the sparsity. We adopt a strategy akin to that employed in supervised learning, incorporating a regularization loss term, e.g. $L_1$ or $L_2$ regularization, and transform the sparsity constraint to the term $w L^{MASK}_t$ in Eqn. (7) and thus set $w$ as a hyper-parameter. Additionally, to investigate the impact of $w$, we vary the hyper-parameter $w$ from {0, 1e-5, 1e-4, 1e-3, 1e-2} and do an ablation study in Supp. S5.3.
> > >
> > >
> > > Regarding your fourth question, we sincerely apologize for any inconvenience caused. It has come to our attention that there was a synchronization issue with the code file in the Supplementary Material. We are sorry that the code in the Supplementary Material is not up-to-date and the old version mainly investigates the alternative design mentioned in Supp. S10. We observe that the alternative design has noticeable fidelity when the agent is near-optimal while performing worse when the agent is sub-optimal. It motivates us to redesign the goal and introduce a Lagrange multiplier to solve the problem. It is worth noting that we implemented the code for optimizing $\lambda$ in two MuJoCo games, namely You-Shall-Not-Pass and Kick-And-Defend, in the old version. You can locate this code in the file ppo2_mask.py, specifically at line 755, within the “normal_form/YouShallNotPass/src” and “normal_form/KickAndDefend/src” folders. Although we are still working on cleaning the code, we are committed to providing accurate and up-to-date resources for our readers and reviewers. We have uploaded the latest version of the code file for Pong and the perfect-information extensive game under the “updated_code” folder. You can access the corrected code from the link in the Supplementary Material. For your convenience, in the "updated_code/Pong" folder, you can find the code for optimizing Eqn. (8) within ppo.py, specifically at line 257. In the "updated_code/perfect_game" folder, the code for optimizing Eqn. (8) can be located within ppo_gmax.py, at line 500. We will release the cleaned code upon publication. We deeply regret any inconvenience this has caused and appreciate your understanding and patience as we rectify this issue. Thank you for bringing this matter to our attention.
> > >
> > >
> > > [4] Constrained Reinforcement Learning Has Zero Duality Gap. In Proc. of NeurIPS, 2019.
> > >
> > > [5] On The Robustness Of Safe Reinforcement Learning Under Observational Perturbations. In Proc. of ICLR, 2023.
> > >
> > > [6] Constrained optimization and Lagrange multiplier methods. Academic press, 2014.

---

### Official Review · Reviewer_oH2u · 2023-07-05

**Soundness:** 3 good
**Presentation:** 3 good
**Contribution:** 3 good
**Rating:** 6
**Confidence:** 4

**Summary:**

This submission focuses on explaining which states are important to the agent’s final reward.
By utilizing a mask to learn and assess which actions are critical. When learning the mask, it focuses on the random actions without affecting the agent’s performance. They evaluate on 10 different tasks such as Pong, some scenarios in StarCraft II, and Connect 4. After learning the mask, they provide fidelity scores and show some examples like Pong and Connect 4. With this information from the mask, they utilize it to perform adversarial attacks and correct agent errors by fine-tuning. Their work outperforms among EDGE, lazy MDP, and value-max.



**Strengths:**

Significance and originality:
How ubiquitous this method can be since as they stated, it does not assume access to the agent’s value function or policy network. Hence it can be utilized by multiple methods as well as individual agents in a multi-agent environment. Figures 4 and 5 are interesting to showcase which time steps are important based on the fidelity scores. These figures showcase a one player game and a two player game showcasing the differences especially in Connect 4. Using the explanations to provide adversarial attacks and correcting agent’s sub-optimal performance is interesting. In Table 1, your implementation outperforms EDGE and others in both performance drop for adversarial attacks and performance gain in patching.

Edit: I have read their rebuttal and I will change the score from a 5 to a 6.

**Weaknesses:**

Evaluation:
More evaluations among other networks to see how versatile it is. Plus other ablations such as do you vary the amount of time steps for the input, like frame stacking to assess how the time step prioritization could affect among the parameter value for frame stacking.

Clarity:
In the supplementary material when reading it, the networks just said if they were CNN, LSTM, MLP, but were they DQN, A2C, and so on. This is important information to assess with your method if it can learn the different masking for them. Plus it would be interesting as another experiment to see if DQN or Double DQN focus on different time steps.

Related Works:
Related works to reference perturbation methods because in the design rationale what you are mentioning is very similar to what computer vision has done with perturbation methods to understand visual explanations with salience maps. You even use the nomenclature perturb like in the second paragraph of design rationale. For instance to include perturbation computer vision methods like RISE (Petsiuk, Vitali, Abir Das, and Kate Saenko. "Rise: Randomized input sampling for explanation of black-box models." arXiv preprint arXiv:1806.07421 (2018).) since it creates perturbation masks and to suggest that there has been work in computer vision using perturbations. Yes, you focus on the time steps but can mention that there has been work that wants to show which pixels are affecting classification. What you are using in your masking approach is still novel for reinforcement learning.

**Questions:**

There were no questions per say just suggestions in the weaknesses section.

**Limitations:**

They do mention challenges with their approach such as converge issues. So they did address some limitations. Other negative societal impacts is not an issue for this submission since they are working on the opposite part where they want to understand for reinforcement learning why decisions are being made.

---

> ### Author Rebuttal · Authors · 2023-08-09
>
> We thank Reviewer oH2u for the constructive and insightful comments. Please see our response to each of your questions below.
>
> **1. Evaluation: More evaluations among other networks to see how versatile it is. Plus other ablations such as do you vary the amount of time steps for the input, like frame stacking to assess how the time step prioritization could affect among the parameter value for frame stacking.**
>
> We appreciate the reviewer's suggestion to perform more evaluations among different networks and conduct further ablation studies. These suggestions indeed provide avenues for enhancing the comprehensiveness of our research.
>
> It is worth noting that we do not assume access to the target agent's network. We report the network structures of StateMask in Supp. S4.2, which vary from CNN to LSTM and MLP. This demonstrates the versatility of our method in accommodating diverse network structures for StateMask. We appreciate your suggestion to explore more evaluations across various networks, and will consider further investigations on other networks such as the Transformer to enhance the comprehensiveness of our evaluation in future research.
>
> Regarding the method of encoding the state (i.e., either considering only the current frame or stacking multiple frames), we conduct an experiment in the Pong game. Specifically, we verify the number of frames stacked for the StateMask input in the range of {1, 2, 4} and see how the fidelity changes. We show the fidelity score comparison in Figure 3(b) of the attached PDF. We can observe that stacking two and four frames share similar fidelity while the fidelity of stacking only one frame is much lower. We suspect the reason is that we need temporal information (e.g., the movement of the ball) when training the StateMask in the Pong game. In contrast, for games (e.g., Connect 4 or Tic-Tac-Toe) where temporal dependencies play a less role, the influence of frame stacking may be less pronounced.
>
> **2. Clarity: In the supplementary material when reading it, the networks just said if they were CNN, LSTM, MLP, but were they DQN, A2C, and so on? This is important information to assess with your method if it can learn the different masking for them. Plus it would be interesting as another experiment to see if DQN or Double DQN focus on different time steps.**
>
> Your suggestion to examine the impact of using different reinforcement learning methods like DQN or A2C to train the mask network is insightful. However, our current design does not permit the direct application of these methods because our method relies on iteratively updating $\theta$ and $\lambda$ by solving Eqn. (7) and Eqn. (8). To utilize DQN or A2C, we have to redesign the objective function that directly maximizes the agent’s expected total reward after applying the state mask. As an additional experiment, we implement DQN and A2C on the Pong game using the new objective function and compare them with our method and report the fidelity scores in Figure 3(a) of the attached PDF. The visualization results in Figure 1, show that the state masks learned by DQN and A2C focus on different time steps from ours and have a lower fidelity compared to our method.
>
> **3. Related Works: Include perturbation methods such as RISE in the computer vision area.**
>
> We appreciate the reviewer's suggestion to reference perturbation methods in the field of computer vision, such as the RISE [1] (e.g., a saliency map method based on randomly masking the inputs and obtaining the corresponding outputs). In light of your suggestion, we will certainly include a discussion of related work in perturbation methods from the computer vision area and compare StateMask with them in the next version. Thank you for bringing this to our attention.
>
> [1] RISE: Randomized input Sampling for Explanation of Black-box Models. In Proc. of BMVC, 2018.

---

> > ### Comment · Reviewer_oH2u · 2023-08-14
> > **Reply to your rebuttal**
> >
> > Thank you for providing additional clarification. It is greatly appreciated. With the additional experiments, I will raise my score from a 5 to a 6.

---

> > > ### Author Response · Authors · 2023-08-14
> > > **Reply to Reviewer oH2u**
> > >
> > > We are genuinely thankful for your insightful comments and we are pleased to see that our efforts have positively impacted your assessment. In our forthcoming version, we plan to include perturbation methods in the field of computer vision in the related work section. Furthermore, we will discuss two alternative designs and the effect of frame stacking. Your feedback continues to be invaluable as we enhance our work.

---

### Official Review · Reviewer_ENqW · 2023-07-07

**Soundness:** 3 good
**Presentation:** 4 excellent
**Contribution:** 2 fair
**Rating:** 6
**Confidence:** 3

**Summary:**

This paper aims to explain deep RL through identifying the critical states at which the action of policy significantly impacts the final reward performance. The main idea is to learn a state-mask, which is modeled as an additional policy, to determine whether to mask original action output by a random one and minimize the performance difference in the meantime. In practice, the authors adopt a trust-region trick to guarantee the monotonic decrease of performance difference and learn the objective in a PPO style. They further apply the explanation of critical states to do adversarial attack and error patch, which exhibits a better performance than baselines.

**Strengths:**

- The paper is clearly written and the presentation in the evaluation part is good.
- The empirical results and the selected examples show the effectiveness of the proposed method on key state explanation.
- The generated explanation is easily compatible with the downstream tasks, e.g., adversarial attack and defense, which lays a foundation for future work.


**Weaknesses:**

- The paper is implicitly built upon a restrictive assumption that there are some specific single timesteps/states that contribute to the final reward significantly in every episode. However, there are other cases that it is a series of actions (may be consecutive or not) that mutually influences the total reward, which is more common in complex environments but hard to be captured by this method.

**Questions:**

- When using thm 1 to derive the objective function in eq.(5), do you remove the first condition in eq.(4)?
- How do you choose the threshold $c$ in eq.(5) since there is a trade-off between better explanation and smaller performance difference?
- How is the fidelity score in fig. 3 computed? The authors mention it is from [1] but in the original paper the lower score indicates the higher fidelity of explanation (the third last row in page 7).

[1] Guo, Wenbo, et al. "Edge: Explaining deep reinforcement learning policies." Advances in Neural Information Processing Systems 34 (2021): 12222-12236.

---

> ### Author Rebuttal · Authors · 2023-08-09
>
> We thank Reviewer ENqW for the constructive and insightful comments. Please see our response to each of your questions below.
>
> **1. The paper is implicitly built upon a restrictive assumption that there are some specific single timesteps/states that contribute to the final reward significantly in every episode. However, there are other cases that it is a series of actions (may be consecutive or not) that mutually influences the total reward, which is more common in complex environments but hard to be captured by this method.**
>
> The reviewer questioned that our paper is based on the assumption about the significant influence of specific timesteps or states on the final reward. However, we would like to clarify that our method, StateMask, does not inherently depend on such an assumption. StateMask is designed to identify not only individual states but also multiple states that contribute towards the final reward. It is capable of tracing back to multiple actions that mutually influence the reward, regardless of consecutive or not.
>
> Evidence of this can be seen in Figure 5 of our paper. Our StateMask method identifies two important non-consecutive action series in the complex game Connect 4. The first critical action series marked by our method is the initial move. This move, as explained by [1], sets the foundation for future gameplay. Furthermore, StateMask highlights the last two time steps as significant contributors to the agent's victory. The second-to-last step stands out as pivotal, as the black player has a winning strategy no matter which column the blue player places their piece next. This showcases StateMask's effectiveness in capturing joint contributions of actions in complex environments. Thus, we believe our method sufficiently addresses the scenario posed by the reviewer and is versatile enough to handle complex environments where multiple action series can mutually affect the total reward.
>
> [1] A knowledge-based approach of connect-four. Journal of the International Computer Games Association, 1988.
>
> **2. When using Theorem 1 to derive the objective function in Eqn. (5), do you remove the first condition in Eqn. (4)?**
>
> No, we do not remove the first condition in Eqn. (4) when using Theorem 1 to derive the objective function in Eqn. (5). In fact, optimizing Eqn. (5) is equivalent to optimizing Eqn. (4). We use the same trick to deal with the first condition in Eqn. (4) as TRPO [2] and transform it to maximize the local approximation $L\_{\tilde{\pi}\_{\theta\_{old}}}(\widetilde{\pi}\_{\theta})$
>  with a trust region constraint. We recommend referring to Supplement S2, where we give a detailed analysis to clarify this point.
>
> [2] Trust region policy optimization. In Proc. of ICML, 2015.
>
> **3. How do you choose the threshold $c$ in Eqn. (5) since there is a trade-off between better explanation and smaller performance difference?**
>
> We appreciate your inquiry about the threshold $c$ in Eqn. (5). While we do not explicitly choose the threshold $c$ in Eqn. (5), we transform the constraint $E\_{a^e \sim \tilde{\pi}\_{\theta}}[a^e] \geq c$ to an additional loss item $L^{mask}$ with coefficient $w$ when optimizing StateMask. Please refer to Supp. S3 for the detailed derivation. Now, $w$ will be utilized to regulate the proportion of the mask. As stated in Supp. S4.2, we search the $w$ in {0, 1e-5, 1e-4, 1e-3, 1e-2} for each environment. Besides, we also study the sensitivity of $w$ in Supp. S5.3. We can observe that when the hyperparameter $w$ differs, the fidelity scores of our explanation method do not vary too much. This suggests that our model is robust to the choice of "$w$", and thus manages the trade-off effectively.
>
> **4. How is the fidelity score in Figure 3 computed? The authors mention it is from [1] but in the original paper the lower score indicates the higher fidelity of explanation (the third last row in page 7).**
>
> We appreciate your query about the fidelity score presented in Figure 3. We apologize if there was any confusion due to the space limit. In our work, we use a similar but slightly modified approach from [3] to calculate the fidelity scores. Unlike [3], in our method, we have introduced a negative sign, thus a higher score represents higher fidelity. We make this adjustment for ease of presentation. The details of our fidelity test can be found in Supp. S5.1. We believe this comprehensive description will provide clarity on the metric we used.
>
> [3] Edge: Explaining deep reinforcement learning policies. In Proc. of NeurIPS, 2021.

---

> > ### Comment · Reviewer_ENqW · 2023-08-15
> >
> > Thanks for your reply. Most of my concerns have been addressed and I have some suggestions on fig.4.
> >
> > I roughly agree that Statemask is capable of capturing non-consecutive key actions but the illustration in fig.4 are not very convincing as the identification of critical steps in fig.4 may be not very hard. So probably you can add some comparison with baselines (e.g., other baseline cannot identify the critical time) or move some illustrations on complex tasks (e.g., figures in supplementary S6) to main text in revision.

---

> > > ### Author Response · Authors · 2023-08-15
> > > **Reply to Reviewer ENqW**
> > >
> > > We appreciate your continued engagement with our manuscript and the acknowledgment of our efforts in addressing your concerns. Your insights are valuable to us in ensuring the quality of our work. Although we are unable to submit further revisions at this stage, your suggestions will undoubtedly be incorporated into the subsequent version. We will move some illustrations on complex tasks (e.g. DouDizhu) from Supplement S6 to the main text and include visualization comparisons with baselines in the Supplementary Material in the revised version. We are grateful for your constructive feedback.

---

### Author Rebuttal · Authors · 2023-08-09

Dear Reviewers,

We would like to express our sincere gratitude for your thoughtful and constructive feedback on our manuscript. Your insights and suggestions have significantly enriched the quality of our work, and we appreciate the time and effort you have dedicated to reviewing our paper.

In response to your valuable recommendations, we have diligently incorporated additional experiments that align with your suggestions. Following the suggestion of Reviewer oH2u, we further investigate two alternative designs DQN and A2C, and the effect of frame stacking in the Pong game. In response to Reviewer AA7o, we do additional experiments on a shortest path finding problem -- MiniGrid-Empty-6x6-v0 and visualize the identified critical path. We have thoughtfully documented these experiments and their outcomes in the attached PDF document, which we believe adds substantial value to the overall contribution of our research.

Your input has been instrumental in shaping the evolution of our paper, and we hope that the additional experiments and results we have provided effectively address your concerns and contribute positively to the overall understanding of our methodology.

Once again, we extend our heartfelt appreciation for your invaluable feedback, which has undoubtedly contributed to the advancement of our research.

---

### Decision · Program_Chairs · 2023-09-21

**Decision:**

Accept (poster)

**Comment:**

This paper seeks to explain the (black-box natured) deep RL methods by identifying the critical states that would impact the final reward. It learns a StateMask network that blinds a target agent and forces it to take random actions at some steps without compromising the agent's performance. All the reviewers agree that the paper is well-written and has novel and significant contribution. The experimental results are solid in showing the effectiveness of the proposed method. Furthermore, the feedbacks provided by all the reviewers have been sufficiently addressed during the rebuttal phase.